# MVAR: Visual Autoregressive Modeling with Scale and Spatial Markovian Conditioning

**Jinhua Zhang**    **Wei Long**    **Minghao Han**    **Weiyi You**    **Shuhang Gu**[*]
University of Electronic Science and Technology of China
jinhua.zjh@gmail.com   shuhanggu@gmail.com

## Abstract

Essential to visual generation is efficient modeling of visual data priors. Conventional next-token prediction methods define the process as learning the conditional probability distribution of successive tokens. Recently, next-scale prediction methods redefine the process to learn the distribution over multi-scale representations, significantly reducing generation latency. However, these methods condition each scale on all previous scales and require each token to consider all preceding tokens, exhibiting scale and spatial redundancy. To better model the distribution by mitigating redundancy, we propose **M**arkovian **V**isual **A**uto**R**egressive modeling (**MVAR**), a novel autoregressive framework that introduces scale and spatial Markov assumptions to reduce the complexity of conditional probability modeling. Specifically, we introduce a scale-Markov trajectory that only takes as input the features of adjacent preceding scale for next-scale prediction, enabling the adoption of a parallel training strategy that significantly reduces GPU memory consumption. Furthermore, we propose spatial-Markov attention, which restricts the attention of each token to a localized neighborhood of size $k$ at corresponding positions on adjacent scales, rather than attending to every token across these scales, for the pursuit of reduced modeling complexity. Building on these improvements, we reduce the computational complexity of attention calculation from $\mathcal{O}(N^2)$ to $\mathcal{O}(Nk)$, enabling training with just eight NVIDIA RTX 4090 GPUs and eliminating the need for KV cache during inference. Extensive experiments on ImageNet demonstrate that MVAR achieves comparable or superior performance with both small model trained from scratch and large fine-tuned models, while reducing the average GPU memory footprint by **3.0×**.

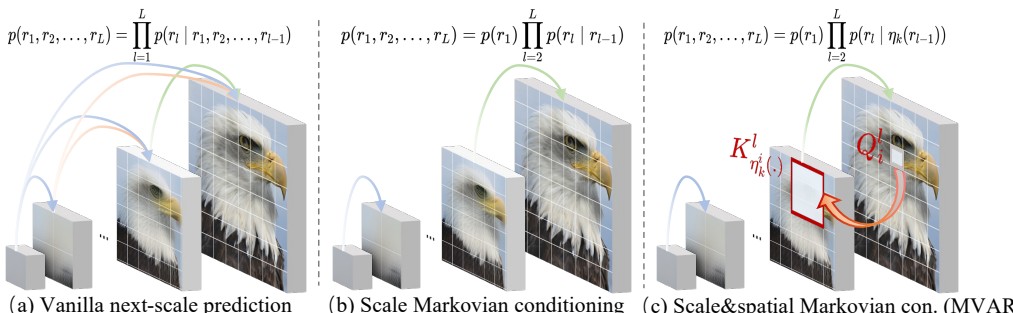

$$p(r_1, r_2, \ldots, r_L) = \prod_{l=1}^{L} p(r_l \mid r_1, r_2, \ldots, r_{l-1}) \qquad p(r_1, r_2, \ldots, r_L) = p(r_1) \prod_{l=2}^{L} p(r_l \mid r_{l-1}) \qquad p(r_1, r_2, \ldots, r_L) = p(r_1) \prod_{l=2}^{L} p(r_l \mid \eta_k(r_{l-1}))$$

(a) Vanilla next-scale prediction    (b) Scale Markovian conditioning    (c) Scale&spatial Markovian con. (MVAR)

Figure 1: **Vanilla next-scale prediction *vs*. MVAR. Left**: The vanilla next-scale method predicts each scale based on all previous scales and requires each token to consider all preceding tokens. **Middle**: A next-scale variant predicts the next scale using only the adjacent scale, leveraging scale Markovian conditioning across scales. **Right**: MVAR further disentangles the spatial constraint and predicts each token using the neighboring positions of adjacent scales, based on spatial Markovian conditioning.

## 1 Introduction

Autoregressive (AR) models (Van Den Oord et al., 2016; Esser et al., 2021; Ramesh et al., 2021; Sun et al., 2024; Li et al., 2024b), inspired by the next-token prediction mechanisms of language models,

---

[*]Corresponding author        **Project page:** https://nuanbaobao.github.io/MVAR

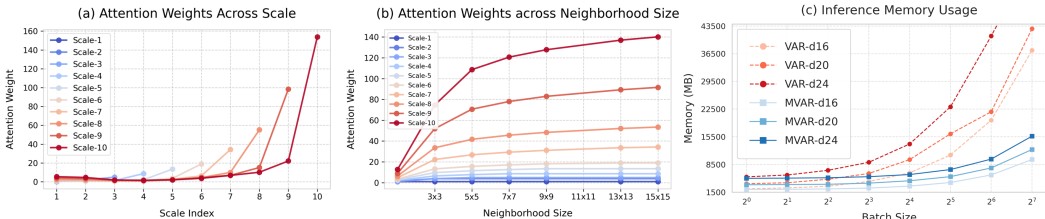

Figure 2: **Left**: Accumulated attention weights across scales in VAR. At each scale, the highest attention weight occurs at the adjacent scale, indicating a predominant focus on neighboring scales rather than all preceding ones. **Middle**: Accumulated attention weights across varying neighborhood sizes between adjacent scales in VAR. As the neighborhood expands, attention weight growth becomes more gradual, indicating a spatially localized focus rather than engagement with all preceding tokens. **Right**: Memory usage during inference. MVAR reduces memory consumption by **4.2×** compared to VAR-$d24$ (9,860MB *vs.* 40,971MB at batch size 64) through scale and spatial Markovian conditioning, demonstrating a significant improvement in memory consumption.

have emerged as a powerful paradigm for modeling visual data priors in image generation tasks. Conventional AR models (Esser et al., 2021; Wang et al., 2024) typically adopt a raster-scan order for conditional probability modeling, which suffers from excessive decoding steps and results in substantial generation latency. To mitigate this latency, pioneering studies have explored modifying token representation patterns (Lee et al., 2022; Huang et al., 2025; Pang et al., 2024; Liu et al., 2024) or redefining the decoding order (Chang et al., 2022; Yu et al., 2023; Li et al., 2024b). Recently, visual autoregressive modeling (VAR) (Tian et al., 2024) has shifted visual data priors modeling from next-token prediction to a next-scale prediction paradigm, which sequentially generates multi-scale image tokens in a coarse-to-fine manner. As the next-scale prediction paradigm better preserves the intrinsic two-dimensional structure of images at each scale, VAR improves both the efficiency of the modeling process and the generation quality compared to conventional AR baselines.

Although vanilla next-scale prediction methods (Tian et al., 2024; Tang et al., 2024; Han et al., 2024; Chen et al., 2024) dramatically accelerate the generation process, our key observations, as shown in Fig. 2 (a) and Fig. 2 (b), reveal that approximate conditional independence, both across scales and between adjacent spatial locations, can be further leveraged to mitigate redundancy and improve the generation process. More specifically, vanilla next-scale prediction methods utilize tokens from all preceding scales to predict tokens of next-scale. However, as illustrated in Fig. 2 (a), the analysis of attention weights across scales reveals that each scale predominantly relies on its immediately preceding neighbor and is approximately independent of non-adjacent scales. Such cross-scale conditional independence allows us to establish scale-Markov next-scale prediction model for reducing GPU memory consumption and eliminating redundant computation. On the other hand, conventional next-scale prediction methods adopt the features of all preceding tokens as keys and values in attention operations (Vaswani, 2017). Nevertheless, as represented in Fig. 2 (b), the analysis of accumulated attention weights across adjacent scales with different neighborhood sizes reveals that each token primarily relies on its immediate neighbors rather than all preceding tokens. Such approximate spatial conditional independence makes it possible to further simplify our scale-Markov model with a spatial-Markov constraint.

Building upon the above observations, we introduce scale and spatial Markovian assumptions into the vanilla next-scale prediction model and propose **M**arkovian **V**isual **A**uto**R**egressive (**MVAR**) model for effective modeling of visual data priors. Instead of relying on all preceding scales to predict the current scale, as shown in Fig. 1 (a) for conventional next-scale prediction method, we introduce scale-Markov assumption which models the transition probabilities between adjacent scales, as illustrated in Fig. 1 (b). This design enables the learning of a scale-Markov trajectory with a parallel training strategy, significantly reducing GPU memory consumption and computational redundancy. In addition to the scale-Markov assumption, we further introduce a spatial-Markov constraint which confines each token's receptive field to a neighborhood of size $k$ at its corresponding position. As depicted in Fig. 1 (c), the proposed spatial-Markov constraint further reduces the computational footprint of our model. Based on the above improvements, MVAR reduces the computational complexity of attention calculation from $\mathcal{O}(N^2)$ to $\mathcal{O}(Nk)$, enables training on eight NVIDIA RTX 4090 GPUs, and eliminates the need for KV cache during inference. Extensive experiments on

ImageNet 256×256 demonstrate the advantages of our model. Compared to the vanilla VAR model, our proposed Markovian constraints not only reduce memory and computational footprints for visual modeling, but also improve modeling accuracy by focusing on more important information.

Our main contributions are summarized as follows:

- We propose a novel next-scale autoregressive framework with a scale Markovian assumption to model conditional likelihood. It substantially reduces memory consumption during both training and inference, enabling MVAR training on eight NVIDIA RTX 4090 GPUs.

- Based on the learned scale-Markov transition probabilities, we introduce spatial-Markov attention to mitigate spatial redundancy across adjacent scales, reducing the computational complexity of attention calculation from $\mathcal{O}(N^2)$ to $\mathcal{O}(Nk)$.

- We conduct comprehensive experiments on both small model trained from scratch and large fine-tuned models. Our method achieves comparable or superior performance to the vanilla next-scale prediction model while reducing memory consumption by **4.2**×, as shown in Fig. 2 (c).

## 2 RELATED WORK

### 2.1 GANS AND DIFFUSION IMAGE GENERATION

GANs and diffusion image generation methods have been extensively investigated for image synthesis. In contrast to AR models, GANs (Goodfellow et al., 2020) learn a mapping from a noise distribution to the high-dimensional image space in a single pass via adversarial training, enabling efficient sampling. Pioneering efforts include the LAPGAN (Denton et al., 2015), which progressively generates images from coarse to fine, and StyleGAN (Sauer et al., 2022), which enables controllable, high-fidelity facial synthesis via unsupervised learning. Further progress has been made by improving architectures (Xu et al., 2022), training strategies (Chen et al., 2020), conditioning mechanisms (Peng & Qi, 2019), and evaluation metrics (Gu et al., 2020), advancing both theoretical understanding and practical applications. More recently, unlike the single-step mapping used in GANs, diffusion models decompose image synthesis into a sequence of gradual denoising steps, thereby enabling stable training and improved sample quality. Early methods relied on pixel-space Markovian sampling (Ho et al., 2020), which incurred high computational cost. Subsequent works improved efficiency through non-Markovian samplers (*e.g.*, DDIM) (Song et al., 2020), ODE-based solvers (*e.g.*, DPM-Solver) (Lu et al., 2022), distillation (Salimans & Ho, 2022), latent-space sampling (*e.g.*, LDM) (Rombach et al., 2022b), and modified architectures (Peebles & Xie, 2023b). While GANs and diffusion models both yield high-quality images, GANs suffer from training instability, diffusion models offer stability but suffer from slow sampling, and autoregressive models provide strong causal interpretability and support more efficient sampling.

### 2.2 AUTOREGRESSIVE IMAGE GENERATION

Autoregressive image generation methods originate from sequence-prediction tasks in natural language processing. By recursively modeling the conditional probability of each element given all previous ones, these models enable stepwise image synthesis. Early work such as PixelCNN (Van den Oord et al., 2016) flattened images into one-dimensional pixel sequences and employed gated convolutional layers to capture causal conditional distributions for image synthesis. To overcome the computational bottleneck of next-pixel prediction, VQ-VAE (Van Den Oord et al., 2017) introduced a discrete representation space via a two-stage training process. Subsequently, modeling the next-token distribution in this discrete latent space became the mainstream approach. VQ-GAN (Esser et al., 2021) further enhanced the discrete representations for next-token prediction by integrating adversarial training and a perceptual loss. MaskGit (Chang et al., 2022) introduced a bidirectional transformer with random masking that generates multiple discrete tokens in parallel, significantly reducing the number of autoregressive inference steps. More recently, VAR (Tian et al., 2024) redefined the autoregressive modeling as a next-scale prediction process, mitigating the difficulty of causal modeling. Nonetheless, despite their promising performance, next-scale methods still suffer from excessive memory consumption and high computational costs during training and inference.

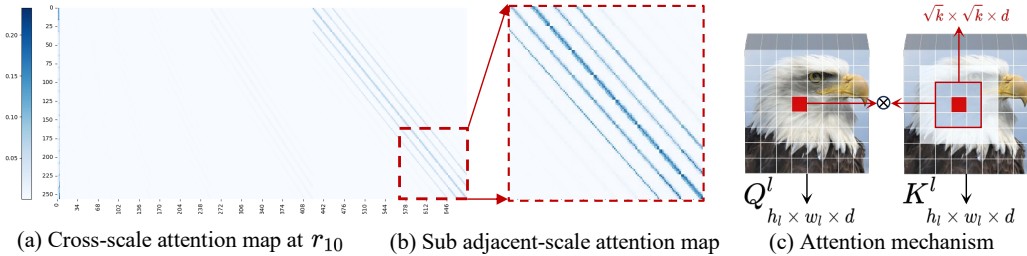

(a) Cross-scale attention map at $r_{10}$    (b) Sub adjacent-scale attention map    (c) Attention mechanism

Figure 3: **Attention patterns in the vanilla next-scale prediction method** (*e.g.*, VAR). **Left**: The vanilla next-scale prediction exhibits redundant inter-scale dependencies, as each scale primarily conditions on its adjacent scale rather than all preceding ones. **Middle**: The attention operation between adjacent scales shows spatial redundancy, as each token mostly attends to its immediate spatial neighbors. **Right**: An illustration of the receptive field in the attention mechanism observed in VAR, showing that attention is restricted to local neighborhoods. Additional visual examples are provided in Appendix F.

## 3 METHODOLOGY

### 3.1 PRELIMINARY: AUTOREGRESSIVE MODELING VIA NEXT-SCALE PREDICTION

The conventional next-scale (Tian et al., 2024; Han et al., 2024) autoregressive modeling introduced by VAR shifts image representation from the raster-scan order token map to multi-scale residual token map, enabling parallel prediction of all token maps at varying scales instead of individual tokens. Following a predefined multi-scale token size sequence $\{(h_l, w_l)\}_{l=1}^{L}$, it first quantizes the feature map $f \in \mathbb{R}^{h \times w \times C}$ into $L$ multi-scale residual token maps $\mathcal{R} = (r_1, r_2, \ldots, r_L)$, each at an increasingly higher resolution $h_l \times w_l$, with $r_L$ matching the resolution of the original feature map $h \times w$. Specifically, the residual token map $\hat{r}_l$ is formulated as:

$$\hat{r}_l = \mathcal{Q}\big(Down\big(f - Up(lookup(Z, r_{l-1}), h, w), h_l, w_l\big)\big), \tag{1}$$

where $\mathcal{Q}(\cdot)$ is a quantizer, $Down(\cdot, (h_l, w_l))$ and $Up(\cdot, (h, w))$ denote down-sampling or up-sampling a token map to the size $(h_l, w_l)$ or $(h, w)$ respectively, and $lookup(Z, r_l)$ refers to taking the $l$-th token map from the codebook $Z$. Then, a standard decoder-only Transformer uses the residual token maps $\mathcal{R}$ to model the autoregressive likelihood:

$$p(r_1, r_2, \ldots, r_L) = \prod_{l=1}^{L} p(r_l \mid r_1, r_2, \ldots, r_{l-1}), \tag{2}$$

where each autoregressive unit $r_l$ is the token map at scale $l$, containing $h_l \times w_l$ tokens. All previous token maps from $r_1$ to $r_{l-1}$, together with the corresponding positional embedding map at scale $l$, serve as the prefill condition for predicting $r_l$. During inference, due to the dense inter-scale connections, where $r_l$ is conditioned on all previous token maps, the KV cache must retain the features of all preceding scales to avoid redundant computation.

### 3.2 KEY OBSERVATIONS

In this subsection, we present the motivation behind MVAR based on two key observations from the vanilla next-scale prediction paradigm (Tian et al., 2024), as revealed by the attention weight analysis in Fig. 2 and the attention map patterns in Fig. 3. First, the attention weights of vanilla next-scale method reveal redundant inter-scale dependencies, leading to excessive GPU memory consumption and computational inefficiency. Second, dense attention operations between adjacent scales display spatial redundancy, where each token primarily attends to its immediate spatial neighbors rather than all preceding tokens, presenting an opportunity to further reduce computational complexity.

***Observation 1: Attention weights in vanilla next-scale prediction model exhibit scale redundancy.***
While vanilla next-scale method utilizes the entire sequence of token maps $(r_1, r_2, \ldots, r_{l-1})$ as the conditioning prefix to predict $r_l$, the information encapsulated in the preceding scale token maps is

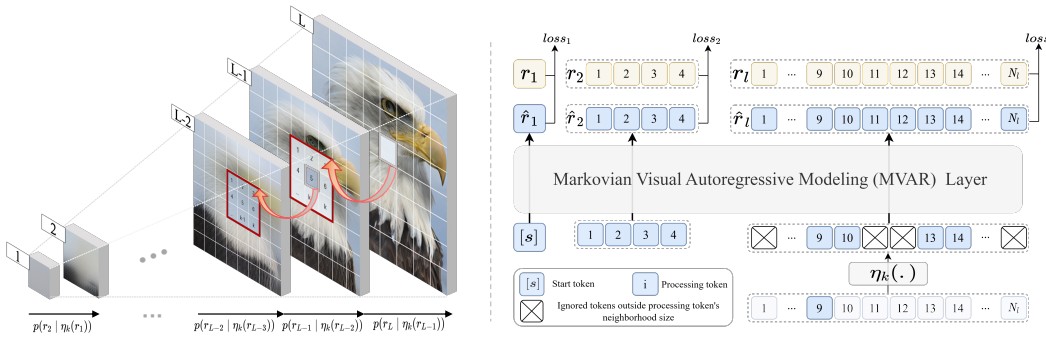

(a) Overall framework of the proposed MVAR

(b) Implementation of scale-Markov trajectory and spatial-Markov attention

Figure 4: **Overall Framework of MVAR. First**, a scale-Markov trajectory predicts $r_l$ using only its adjacent scale $r_{l-1}$, discarding all earlier scales. This allows for parallel training across scales using a standard cross-entropy loss $loss_l$. **Second**, a spatial-Markov attention mechanism restricts attention to a local neighborhood of size $k$, reducing computational complexity from $\mathcal{O}(N^2)$ to $\mathcal{O}(Nk)$.

often redundant for consecutive scale prediction. As demonstrated in Fig. 2 (a) and Fig. 3 (a), the attention weights across scales reveal that queries at scale $l$ pay negligible attention to all preceding scales, but exhibit a significant concentration on the immediately adjacent scale. This behavior aligns with the hierarchical design, where higher-resolution scales prioritize local refinements rather than reusing features from coarser levels. Consequently, by effectively alleviating such scale redundancy, we can reduce computational complexity and eliminate unnecessary conditioning on earlier scales when predicting next-scale tokens.

*Observation 2: Attention weights in vanilla next-scale prediction model exhibit spatial redundancy.* Conventional next-scale autoregressive approaches typically utilize Transformer (Radford et al., 2019) blocks to model transition probabilities. At the core of Transformer is the attention operation, which enhances features by computing weighted sums of token similarities. This operation is formally defined as:

$$Attention(\boldsymbol{Q}^l, \boldsymbol{K}^l, \boldsymbol{V}^l) = SoftMax\left(\boldsymbol{Q}^l(\boldsymbol{K^l})^T/\sqrt{d}\right)\boldsymbol{V}^l, \tag{3}$$

where $\boldsymbol{Q}^l \in \mathbb{R}^{N_l \times d}$, $\boldsymbol{K}^l \in \mathbb{R}^{N_l \times d}$, and $\boldsymbol{V}^l \in \mathbb{R}^{N_l \times d}$ are linearly projected from the features of token map $r_l$, $N_l = h_l \times w_l$ is the token number, and $d$ is feature dimension. While attention operations theoretically enable global context modeling, finer scales which primarily capture textural details exhibit spatial redundancy, such as strong locality and 2D spatial organization, akin to convolutional operations (He et al., 2016). As depicted in Fig. 3 (b), the attention map between adjacent scales exhibits a *diagonally dominant* pattern, indicating that adjacent scales interact predominantly through spatially correlated regions rather than global dispersion. Such geometrically constrained weight distribution mirrors the local connectivity prior inherent in convolutional operations, as illustrated in Fig. 2 (b) and Fig. 3 (c). As neighborhood size increases, attention distributions become smoother, implying that attention operations across adjacent scales emphasize local textural details over global structural information (Chen et al., 2024; Guo et al., 2025; Kumbong et al., 2025). Therefore, such spatial redundancy can be leveraged to significantly reduce the $\mathcal{O}(N_l^2)$ time complexity of attention computation, particularly pronounced at finer scales (*e.g.*, $r_9$ and $r_{10}$), where attention operations account for up to 60% of the total computational cost.

In summary, the key insights from our observations are as follows: 1) Redundant inter-scale dependencies in hierarchical architectures can be mitigated by introducing the scale-Markovian assumption, which reduces memory usage and enhances computational efficiency. 2) Spatial redundancy in adjacent-scale interactions enables efficient spatial-Markov attention, which approximates localized 2D operations rather than global attention patterns.

### 3.3 Autoregressive Modeling via Scale and Spatial Markovian Conditioning

To improve the modeling of visual data priors, we introduce MVAR, a novel next-scale autoregressive framework that applies the scale Markovian assumption to inter-scale dependencies and the spatial

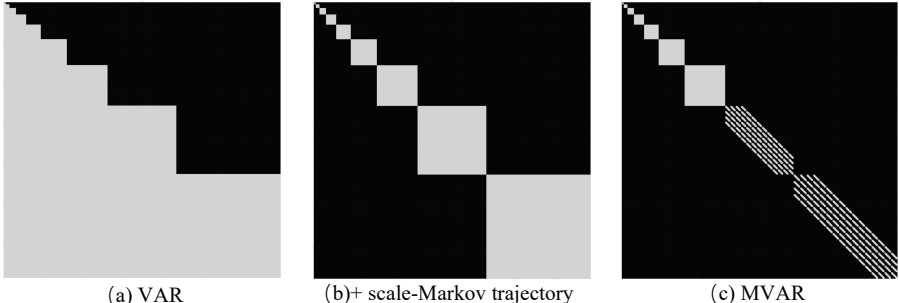

(a) VAR        (b)+ scale-Markov trajectory       (c) MVAR

Figure 5: **Schematic of causal masks for 256×256 image generation.** VAR employs a full causal mask to model $p\big(r_l \mid r_{<l}\big)$. In MVAR, scales $r_1$ to $r_8$ use a **diagonal-pattern mask** to model $p\big(r_l \mid r_{l-1}\big)$, since the receptive field is smaller than the neighborhood size $k$. Scales $r_9$ and $r_{10}$ are generated using custom CUDA kernels to model $p\big(r_l \mid \eta_k(r_{l-1})\big)$.

Markovian assumption to attention across adjacent scales, as illustrated in Fig. 4. In this framework, we restrict inter-scale dependencies to adjacent scales and incorporate localized attention operations into the dense computations between them, reducing memory consumption and computational cost.

**Next-scale prediction with scale Markovian conditioning.** To better mitigate the observed redundant dependencies across scales, we reformulate autoregressive modeling for next-scale prediction by shifting from the "previous scales-based" paradigm to an "adjacent scale-based" paradigm. Specifically, the conditioning prefix for scale $r_l$ is restricted to its immediate predecessor $r_{l-1}$, discarding all other preceding scales. Formally, the autoregressive transition probability is defined as:

$$p(r_1, r_2, \ldots, r_L) = p(r_1) \prod_{l=2}^{L} p\big(r_l \mid \eta_k(r_{l-1})\big), \tag{4}$$

where $\eta_k(\cdot)$ restricts attention to preceding tokens within a neighborhood of size $k$, as discussed below. The term $p(r_1)$ represents the probability distribution of the first scale token, generated from a class embedding referred to as the start token $[s]$. This pruning strategy aligns with the scale redundancy observed in Fig. 2 (a) and Fig. 3 (a), allowing each scale to focus on local refinements rather than redundantly reusing information from preceding scales.

**Attention operation with spatial Markovian conditioning.** As illustrated in Fig. 1 (b), restricting inter-scale dependencies to adjacent scales reduces redundant interactions; however, dense computations within these adjacent scales remain excessive (see analysis in Fig. 2 (b) and attention map patterns in Fig. 3 (b)). As shown in Eq. 3, conventional attention operates densely over all tokens in adjacent scales to produce outputs, resulting in quadratic time complexity $\mathcal{O}(N_l^2)$. To mitigate the observed spatial redundancy, we introduce spatial-Markov attention (Hassani & Shi, 2022; Hassani et al., 2023), where each query token attends exclusively to its $k$ nearest neighbors, as illustrated in Fig. 4. Specifically, given the query, key, and value matrices $\boldsymbol{Q}^l \in \mathbb{R}^{N_l \times d}$, $\boldsymbol{K}^l \in \mathbb{R}^{N_l \times d}$, and $\boldsymbol{V}^l \in \mathbb{R}^{N_l \times d}$ at scale $r_l$, we compute the local attention score $\boldsymbol{S}_i^l \in \mathbb{R}^{1 \times k}$ for the $i$-th token within a neighborhood of size $k$ as:

$$\boldsymbol{S}_i^l = \begin{bmatrix} \boldsymbol{Q}_i^l (\boldsymbol{K}_{\eta_k^i(1)}^l)^T & \boldsymbol{Q}_i^l (\boldsymbol{K}_{\eta_k^i(2)}^l)^T & \cdots & \boldsymbol{Q}_i^l (\boldsymbol{K}_{\eta_k^i(k)}^l)^T \end{bmatrix}, \tag{5}$$

where $\eta_k^i(j)$ denotes the index of the $j$-th neighboring token. Similarly, the local value matrix $\boldsymbol{V}_i^l \in \mathbb{R}^{k \times d}$ is defined as:

$$\boldsymbol{V}_i^l = \begin{bmatrix} (\boldsymbol{V}_{\eta_k^i(1)}^l)^T & (\boldsymbol{V}_{\eta_k^i(2)}^l)^T & \cdots & (\boldsymbol{V}_{\eta_k^i(k)}^l)^T \end{bmatrix}^T. \tag{6}$$

Since each token attends only to its $k$ neighboring tokens rather than all tokens as in conventional next-scale paradigms, the computational complexity is reduced from $\mathcal{O}(N_l^2)$ to $\mathcal{O}(N_l k)$. Finally, the spatial-Markov attention output $SA_i^l \in \mathbb{R}^{1 \times d}$ for the $i$-th token with neighborhood size $k$ is defined as:

$$SA_i^l = SoftMax(\boldsymbol{S}_i^l / \sqrt{d}) \, \boldsymbol{V}_i^l. \tag{7}$$

Table 1: **Quantitative results on class-conditional ImageNet at resolution 256×256.** ↓ / ↑ indicate that lower / higher values are better. We report results for representative generative models including generative adversarial networks (GANs), diffusion models (DMs), token-wise autoregressive models, and scale-wise autoregressive models. Metrics marked as "–" are not reported in the original papers. VAR is evaluated using the official pretrained weights from its GitHub repository. "$d16$" denotes the depth of the Transformer in the autoregressive network.

| Type | Models | FID↓ | IS↑ | Precision↑ | Recall↑ | #Params | #Steps |
|---|---|---|---|---|---|---|---|
| GANs | BigGAN (Brock et al., 1809) | 6.95 | 224.5 | 0.89 | 0.38 | 112M | 1 |
| | GigaGAN (Kang et al., 2023) | 3.45 | 225.5 | 0.84 | 0.61 | 569M | 1 |
| | StyleGAN-XL (Sauer et al., 2022) | 2.30 | 265.1 | 0.78 | 0.53 | 166M | 1 |
| DMs | ADM (Dhariwal & Nichol, 2021) | 10.94 | 101.0 | 0.69 | 0.63 | 554M | 250 |
| | CDM (Ho et al., 2022) | 4.88 | 158.7 | – | – | – | 8100 |
| | LDM-4-G (Rombach et al., 2022a) | 3.60 | 247.7 | – | – | 400M | 250 |
| | Simple-Diffusion (Hoogeboom et al., 2023) | 2.44 | 256.3 | – | – | 2B | – |
| | DiT-L/2 (Peebles & Xie, 2023a) | 5.02 | 167.2 | 0.75 | 0.57 | 458M | 250 |
| | DiT-XL/2 (Peebles & Xie, 2023a) | 2.27 | 278.2 | 0.83 | 0.57 | 675M | 250 |
| Token-wise | MaskGIT (Chang et al., 2022) | 6.18 | 182.1 | 0.80 | 0.51 | 227M | 8 |
| | RCG (cond.) (Li et al., 2024a) | 3.49 | 215.5 | – | – | 502M | 20 |
| | VQVAE-2† (Razavi et al., 2019) | 31.11 | ∼45 | 0.36 | 0.57 | 13.5B | 5120 |
| | VQGAN† (Esser et al., 2021) | 18.65 | 80.4 | 0.78 | 0.26 | 227M | 256 |
| | VQGAN (Esser et al., 2021) | 15.78 | 74.3 | – | – | 1.4B | 256 |
| | VQGAN-re (Esser et al., 2021) | 5.20 | 280.3 | – | – | 1.4B | 256 |
| | ViTVQ (Yu et al., 2021) | 4.17 | 175.1 | – | – | 1.7B | 1024 |
| | ViTVQ-re (Yu et al., 2021) | 3.04 | 227.4 | – | – | 1.7B | 1024 |
| | DART-AR (Gu et al., 2025) | 3.98 | 256.8 | – | – | 812M | – |
| | RQTran. (Lee et al., 2022) | 7.55 | 134.0 | – | – | 3.8B | 68 |
| | RQTran.-re (Lee et al., 2022) | 3.80 | 323.7 | – | – | 3.8B | 68 |
| Scale-wise | VAR-$d16$ (Tian et al., 2024) | 3.55 | 280.4 | 0.84 | 0.51 | 310M | 10 |
| | MVAR-$d16$ | 3.09 | 285.5 | 0.85 | 0.51 | 310M | 10 |

Note that during training, by disentangling dependencies between non-adjacent scales such that each $r_l$ only attends to its immediate predecessor $r_{l-1}$, parallelized training can be adopted on a per-scale basis. Specifically, for 256×256 image generation, we maximize GPU utilization by training scales $r_1$ through $r_8$ in parallel. This is achieved through a *diagonal-pattern* causal mask (see Fig. 5), which constrains each $r_l$ only attends to its prefix $r_{l-1}$. Subsequently, scales $r_9$ and $r_{10}$ are separately generated using custom CUDA kernels (Hassani & Shi, 2022) to model $p(r_l \mid \eta_k(r_{l-1}))$. During inference, our method eliminates the need for KV cache, thereby further streamlining computational efficiency. More details about the mixed training strategy are provided in Appendix D.5.

## 3.4 ANALYSIS OF COMPUTATIONAL COMPLEXITY

Conventional next-scale paradigms incur high computational cost because the number of tokens at each scale equals the sum of tokens from all preceding scales, and dense computation is performed between adjacent scales. In the token generation process of conventional next-scale autoregressive models, the time complexity of producing $N$ tokens is $\mathcal{O}(N^2)$ (see Appendix A for more details), whereas our method reduces this to $\mathcal{O}(Nk)$. Specifically, consider a predefined sequence of multi-scale tokens $\{(h_l, w_l)\}_{l=1}^{L}$. For simplicity, let $h_l = w_l = \sqrt{N_l}$ for all $1 \le l \le L$, and denote $\sqrt{N} = h = w$. We further define $\sqrt{N_l} = a^{l-1}$ for some constant $a > 1$, such that $a^{L-1} = \sqrt{N}$. At the $l$-th autoregressive generation stage, the total number of tokens is $N_l = a^{2(l-1)}$. Therefore, the time complexity for this stage is $\mathcal{O}(N_l k)$. Summing over all scales yields:

$$\sum_{l=1}^{L} a^{2(l-1)} k = k \frac{a^{2L} - 1}{a^2 - 1} = k \frac{a^2 N - 1}{a^2 - 1} \sim \mathcal{O}(Nk). \tag{8}$$

## 4 EXPERIMENTAL RESULTS

In this section, we present the implementation details, report the main results on the ImageNet 256×256 generation benchmark, and conduct ablation studies to evaluate the contribution of each

Table 2: **Pre-trained *vs.* Fine-tuning Comparison**. Quantitative comparison between pre-trained VAR baselines and our fine-tuned variants (MVAR[†]). Metrics include inference time (seconds per batch), GFLOPs in attention blocks, KV cache footprint (MB), GPU memory usage (MB), training speed (seconds per iteration), and training memory consumption (MB). All experiments were conducted on NVIDIA RTX 4090 GPU with batch size 32. OOM denotes out of GPU memory.

| Methods | Time (s)↓ | GFLOPs↓ | KV Cache↓ | Memory↓ | Train Speed↓ | Train Memory↓ | FID↓ | IS↑ | Precision↑ | Recall↑ |
|---|---|---|---|---|---|---|---|---|---|---|
| VAR-$d16$ | 0.34 | 43.61 | 5704M | 10882M | 0.99 | 34319M | 3.55 | 280.4 | 0.84 | 0.51 |
| MVAR-$d16$[†] | 0.27 | 35.44 | **0** | 3846M (**2.8×**) | 0.61 (**1.6×**) | 20676M | 3.40 | 297.2 | 0.86 | 0.48 |
| VAR-$d20$ | 0.52 | 81.52 | 8500M | 16244M | 1.35 | 48173M | 2.95 | 302.6 | 0.83 | 0.56 |
| MVAR-$d20$[†] | 0.45 | 68.75 | **0** | 5432M (**3.0×**) | 0.79 (**1.7×**) | 27665M | 2.87 | 295.3 | 0.86 | 0.52 |
| VAR-$d24$ | 0.81 | 136.63 | 12240M | 23056M | – | OOM | 2.33 | 312.9 | 0.82 | 0.59 |
| MVAR-$d24$[†] | 0.71 | 118.25 | **0** | 7216M (**3.2×**) | 0.91 | 38579M | 2.23 | 300.1 | 0.85 | 0.56 |

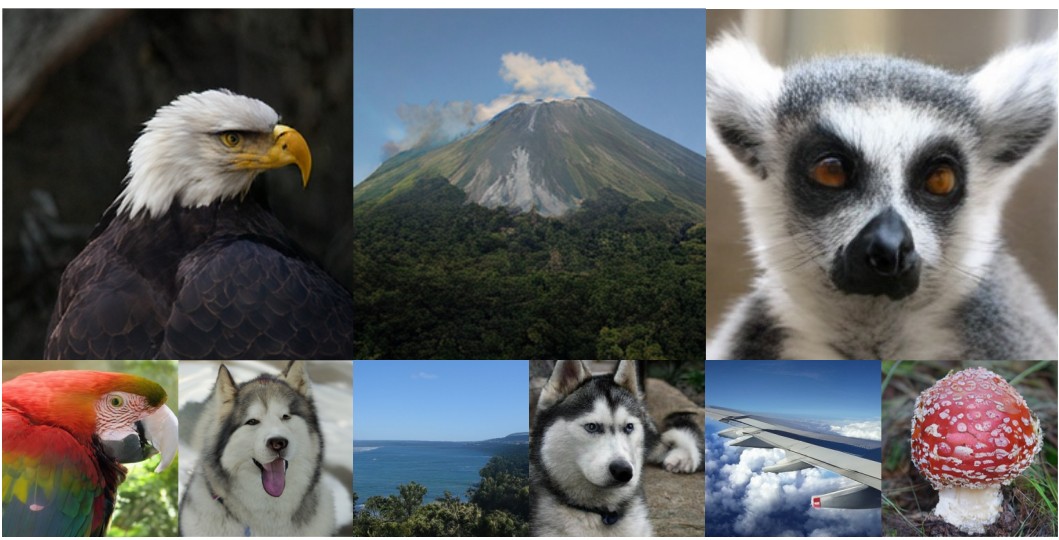

Figure 6: **Qualitative results of MVAR**. Examples of class-conditional generation on ImageNet 256×256.

component in our proposed methodology. Additional experiments on other resolutions and datasets are provided in Appendix D.

### 4.1 IMPLEMENTATION DETAILS

As shown in Tab. 1, we train MVAR-$d16$ model from scratch on eight NVIDIA RTX 4090 GPUs with a total batch size of 448 for 300 epochs. For the results reported in Tab. 2, fine-tuning is performed with a different batch size for 80 epochs. Both models are optimized using the AdamW (Kingma & Ba, 2014) optimizer with $\beta_1 = 0.9$, $\beta_2 = 0.95$, a weight decay of 0.05, and a base learning rate of $1 \times 10^{-4}$. For the predefined multi-scale token length sequence, unless otherwise specified, we adopt the same setting as VAR, i.e., $\{1, 2, 3, 4, 5, 6, 8, 10, 13, 16\}$, for the 256×256 resolution. Following common practice, we evaluate Fréchet Inception Distance (FID) (Heusel et al., 2017), Inception Score (IS), Precision, and Recall, along with additional reference metrics. More implementation details are provided in Appendix B.

### 4.2 MAIN RESULTS

**Overall comparison.** We evaluate the performance of MVAR against the original VAR models, conventional next-token prediction models, diffusion models, and GAN-based approaches on the ImageNet-1K benchmark. As shown in Tab. 1, our method consistently outperforms nearly all competing models, achieving notable improvements over the VAR baselines. Specifically, compared to VAR, our method reduces the FID by 0.46, and increases the IS by 5.1, demonstrating superior overall

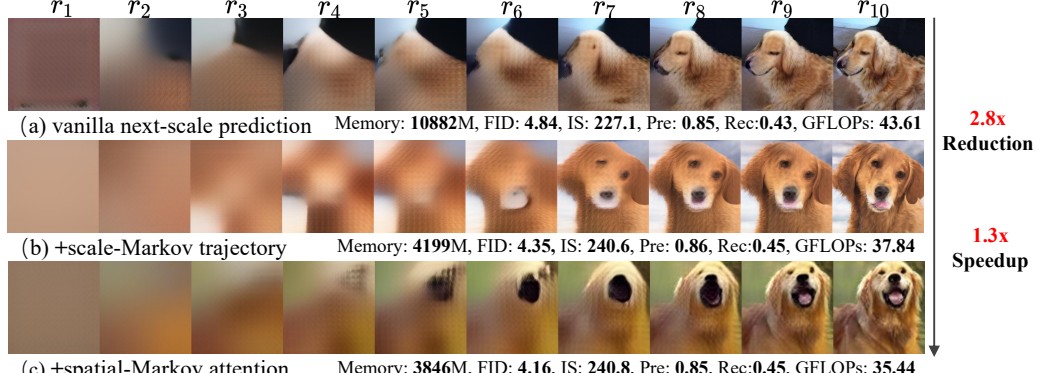

Figure 7: **Ablation study on scale and spatial Markovian conditioning.** Intermediate outputs at scales $r_1$ to $r_{10}$ illustrating the progressive emergence of semantic and spatial details, along with memory usage and evaluation metrics including FID, IS, Precision (Pre), Recall (Rec), and attention GFLOPs. All experiments were conducted on an NVIDIA RTX 4090 with batch size 32.

performance. As illustrated in Fig. 6, MVAR produces high-quality images for class-conditional generation on ImageNet 256×256. Additional qualitative results can be found in Appendix F.

**Pre-trained *vs*. Fine-tuning.** To further demonstrate the scale and spatial redundancy observed in conventional next-scale prediction methods, we use pre-trained VAR models as baselines and then fine-tune them, achieving comparable performance with lower GPU memory consumption and computational costs. Specifically, as shown in Tab. 2, the fine-tuned VAR model not only achieves significant performance improvements over the original model but also attains an average 3.0× reduction in inference memory consumption. These results highlight that our approach outperforms the next-scale paradigm, underscoring the benefits of scale and spatial Markovian conditioning.

### 4.3 ABLATION STUDY

To assess the effectiveness of the core components in our method, we conduct ablation studies focusing on two key aspects: scale Markovian conditioning across scales and spatial Markovian conditioning across adjacent locations. Due to limited computational resources, all results are reported using the 310M model trained under a shorter schedule of 50 epochs.

**Effects of scale and spatial Markovian conditioning.** To assess the impact of key components in MVAR, we design three model variants and compare their performance, as illustrated in Fig. 7. Method (a) serves as the baseline following the VAR setup. To reveal the redundancy of inter-scale dependencies in this setting, Method (b) uses only the immediate predecessor to model transition probabilities, rather than relying on all preceding non-adjacent scales. Method (c) further introduces spatial-Markov attention, refining the attention mechanism to more effectively capture local textural details. As shown in Fig. 7, even when the strong scale and spatial dependencies are removed, the progressive generation process remains analogous to the conventional next-scale prediction method. Moreover, the combined application of scale-Markov trajectory and spatial-Markov attention achieves improvements across most metrics, notably reducing FID by 0.68, increasing IS by 13.7, and lowering GFLOPs of attention operations by 8.17. Additional qualitative results can be found in Appendix F.

**Effects of varying the number of scales in the conditioning prefix.** We conduct experiments to investigate the impact of different strategies for modeling inter-scale dependencies. Method (a) follows the original VAR setup and serves as the baseline. Methods (b) and (c) explore limiting the conditioning prefix to only the two or three most recent preceding scales. As shown in Tab. 3, reducing redundant dependencies leads to lower memory consumption and computational cost. Notably, Method (d), which conditions solely on the immediate predecessor, removes the need for KV cache, improves IS by 13.5, and reduces memory usage by 2.6×. These results suggest that minimizing inter-scale dependencies helps the model concentrate on essential generative patterns rather than redundant historical information, thereby enhancing generation quality.

Table 3: **Ablation study on the number of preceding scales**. Reducing the number of preceding scales decreases memory usage and improves computational efficiency. "(a)" denotes the baseline configuration following the VAR setup. All experiments used an RTX 4090 with batch size 32.

| Method | Number | Memory↓ | KV Cache↓ | Time (s)↓ | GFLOPs↓ | FID↓ | IS↑ | Precision↑ | Recall↑ |
|--------|--------|---------|-----------|-----------|---------|------|------|------------|---------|
| (a) | – | 10882M | 5704M | 0.34 | 43.61 | 4.84 | 227.1 | 0.85 | 0.43 |
| (b) | 3 | 9518M | 3565M | 0.32 | 41.54 | 4.86 | 220.3 | 0.86 | 0.43 |
| (c) | 2 | 9262M | 2147M | 0.31 | 40.15 | 5.01 | 208.8 | 0.84 | 0.45 |
| (d) | 1 | 4199M (**2.6×**) | **0** | 0.29 | 37.84 | 4.35 | 240.6 | 0.86 | 0.45 |

Table 4: **Ablation study on neighborhood size $k$ in spatial-Markov attention.** "–" denotes the baseline configuration following the VAR setup. Additional experiments on other resolutions and on CelebA and FFHQ datasets are provided in Appendix D.4.

| $k$ | FID↓ | IS↑ | Precision↑ | Recall↑ | GFLOPs↓ |
|-----|------|------|------------|---------|---------|
| – | 4.84 | 227.1 | 0.85 | 0.43 | 43.61 |
| 3×3 | 4.64 | 243.9 | 0.85 | 0.44 | 34.89 |
| 5×5 | 4.36 | 235.8 | 0.86 | 0.43 | 35.11 |
| 7×7 | 4.16 | 240.8 | 0.85 | 0.45 | 35.44 |
| 9×9 | 4.18 | 237.4 | 0.85 | 0.46 | 35.89 |

**Effects of neighborhood size $k$ in spatial-Markov attention.** The neighborhood size $k$ is crucial in spatial-Markov attention, as it determines the size of the receptive field. A larger neighborhood provides broader context, generally improving performance. We conduct experiments to examine the effect of varying $k$ from $3 \times 3$ to $9 \times 9$, as shown in Tab. 4. The model improves significantly when $k$ increases to $7 \times 7$, but further increases results in only marginal gains. This is because spatial-Markov attention effectively captures fine-grained details within a compact neighborhood, while larger $k$ values bring diminishing returns and redundant computation, as illustrated in Fig. 2 (b). To strike a balance between performance and computational cost, we choose $k = 7 \times 7$ for our final models.

## 5 CONCLUSION

In this paper, we propose MVAR, a novel next-scale prediction autoregressive framework that significantly reduces GPU memory usage during training and inference while maintaining superior performance. We introduce the scale-Markov trajectory, which models the autoregressive likelihood using only adjacent scales and discards non-adjacent ones. This design enables parallel training and substantially lowers GPU memory consumption. Furthermore, to address spatial redundancy in attention across adjacent scales, we propose spatial-Markov attention, which restricts attention to a neighborhood of size $k$, reducing computational complexity of attention calculation from $\mathcal{O}(N^2)$ to $\mathcal{O}(Nk)$. Extensive experiments demonstrate that our method achieves comparable or better performance than existing approaches while significantly reducing average GPU memory usage.

## ACKNOWLEDGEMENT

This work was supported by National Natural Science Foundation of China (No. 62476051) and Sichuan Natural Science Foundation (No. 2024NSFTD0041).

## REPRODUCIBILITY STATEMENT

We provide detailed hyperparameter settings in Section 4 and Appendix B. To further facilitate reproducibility, we will release our implementation and trained model checkpoints, enabling the reported results to be reproduced.

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

TABLE OF CONTENT FOR APPENDIX

## A  TIME COMPLEXITY ANALYSIS OF DIFFERENT AUTOREGRESSIVE MODELS

In this section, we demonstrate that the time complexities of conventional next-token prediction methods and vanilla next-scale prediction methods (Tian et al., 2024) are $\mathcal{O}(N^3)$ and $\mathcal{O}(N^2)$, respectively. In contrast, MVAR achieves a time complexity of $\mathcal{O}(Nk)$, as established in Section 3.4.

Consider a **standard next-token prediction** using a self-attention transformer, where the total number of tokens is $N = h \times w$. At generation step $i$ ($1 \leq i \leq N$), the model computes attention scores between the current token and all previous $i - 1$ tokens. This operation necessitates $\mathcal{O}(i^2)$ time due to the quadratic complexity of self-attention at each step. Consequently, the overall time complexity is given by

$$\sum_{i=1}^{N} i^2 = \frac{1}{6}N(N+1)(2N+1) \sim \mathcal{O}(N^3). \tag{9}$$

On the other hand, consider a predefined multi-scale length $\{(h_l, w_l)\}_{l=1}^{L}$. For simplicity, let $h_l = w_l = \sqrt{N_l}$ for all $1 \leq l \leq L$, and denote $\sqrt{N} = h = w$. We further define $\sqrt{N_l} = a^{l-1}$ for some constant $a > 1$ such that $a^{L-1} = \sqrt{N}$. For **vanilla next-scale prediction**, at the $l$-th scale generation stage, the total number of tokens can be expressed as

$$\sum_{i=1}^{l} N_i = \sum_{i=1}^{l} a^{2(i-1)} = \frac{a^{2l} - 1}{a^2 - 1}. \tag{10}$$

Thus, the time complexity for the $l$-th scale generation process is $\left(\frac{a^{2l}-1}{a^2-1}\right)^2$. By summing across all $L$ scale generation processes, we obtain the overall time complexity:

$$\sum_{l=1}^{L} \left(\frac{a^{2l}-1}{a^2-1}\right)^2 = \frac{1}{(a^2-1)^2}\left(\frac{a^4(N^2a^4-1)}{a^4-1} + \frac{2a^2(Na-1)}{a^2-1} + \frac{1}{2}\log a^N\right) \sim \mathcal{O}(N^2). \tag{11}$$

Table 5: **Detailed hyper-parameters for our MVAR models.**

| config | value |
| --- | --- |
| *training hyper-parameters* | |
| optimizer | AdamW (Kingma & Ba, 2014) |
| base learning rate | 1e-4 |
| weight decay | 0.05 |
| optimizer momentum | (0.9, 0.95) |
| batch size | 448 ($d16$) / 192 ($d20$) / 384 ($d24$) |
| total epochs | 300 (train) / 80 (fine-tune) |
| warmup epochs | 60 (train) / 2 (fine-tune) |
| precision | float16 |
| max grad norm | 2.0 |
| dropout rate | 0.1 |
| class label dropout rate | 0.1 |
| params | 310M ($d16$) / 600M ($d20$) / 1.0B ($d24$) |
| embed dim | 1024 ($d16$) / 1280 ($d20$) / 1536 ($d24$) |
| attention head | 16 ($d16$) / 20 ($d20$) / 24 ($d24$) |
| *sampling hyper-parameters* | |
| cfg guidance scale (Ho & Salimans, 2022) | 2.7 ($d16$) / 1.5 ($d20$) / 1.4 ($d24$) |
| top-k | 1200 ($d16$) / 900 ($d20$) / 900 ($d24$) |
| top-p | 0.99 ($d16$) / 0.96 ($d20$) / 0.96 ($d24$) |

**For our MVAR method**, at the $l$-th autoregressive generation stage, the total number of tokens is $N_l = a^{2(l-1)}$. Therefore, the time complexity for this stage is $\mathcal{O}(N_l k)$. Summing over all scales yields:

$$\sum_{l=1}^{L} a^{2(l-1)} k = k \frac{a^{2L} - 1}{a^2 - 1} = k \frac{a^2 N - 1}{a^2 - 1} \sim \mathcal{O}(Nk). \tag{12}$$

## B  ADDITIONAL IMPLEMENTATION DETAILS OF MVAR

We list the detailed training and sampling hyper-parameters for MVAR models in Table 5. For the predefined scale lengths inherited from VAR (Tian et al., 2024) $(1, 2, 3, 4, 5, 6, 8, 10, 13, 16)$, we parallelize training across scales $r_1$–$r_8$ to maximize GPU utilization, employing **diagonal-pattern** causal masks. This setup yields a uniform total token length of 255, which closely matches the token lengths of the final two scales ($13 \times 13$ and $16 \times 16$). In addition, we describe the procedures used to visualize Fig. 2 and Fig. 3 for completeness. For Fig. 2(a), we first use VAR to generate the ImageNet validation set, producing one image for each of the 1,000 classes. We then compute the mean attention weights across all heads and layers at scale $r_l$ with respect to the preceding scales $r_1, r_2, \ldots, r_{l-1}$. For Fig. 2(b), we aggregate the attention weights from $p(r_l \mid r_{l-1})$ at each position $(i, j)$ and average them across different neighborhood sizes $k$. For Fig. 3, we again use VAR to generate the ImageNet validation set and visualize the mean attention weights across all layers and heads at the final scale $r_{10}$.

## C  LIMITATIONS AND FUTURE WORK

In this work, we focus on the learning paradigm within the autoregressive transformer framework while retaining the original VQ-VAE architecture from VAR. We anticipate that integrating more advanced image tokenizers (Han et al., 2024; Jiao et al., 2025) could further enhance autoregressive visual generation performance. Future work may explore high-resolution image generation and video synthesis, which remain prohibitively expensive for traditional methods due to KV cache memory demands. Unlike conventional autoregressive models that rely on KV cache, our approach is inherently KV cache-free, reducing both memory consumption and computational complexity.

Table 6: **Quantitative results on ImageNet-512×512**. All experiments were evaluated with a batch size of 64 during inference and a batch size of 6 during training, using an RTX 4090 GPU.

| Method | Memory↓ | KV Cache↓ | Time↓ | GFLOPs↓ | Train speed↓ | Train Memory↓ | FID↓ | IS↑ | Precision↑ | Recall↑ |
|---|---|---|---|---|---|---|---|---|---|---|
| VAR-$d16$ | 43826M | 18790M | 1.98s | 219.88 | 1.26s | 41944M | 8.03 | 187.1 | 0.83 | 0.26 |
| MVAR-$d16$ | 15090M (**2.9x**) | 0 | 1.69s | 116.37 | 0.56s (**2.3x**) | 19510M (**2.1x**) | 7.55 | 187.9 | 0.84 | 0.26 |

VAR (Memory=43826M, FID=8.03)    **2.9x**    MVAR (Memory=15090M, FID=7.55)

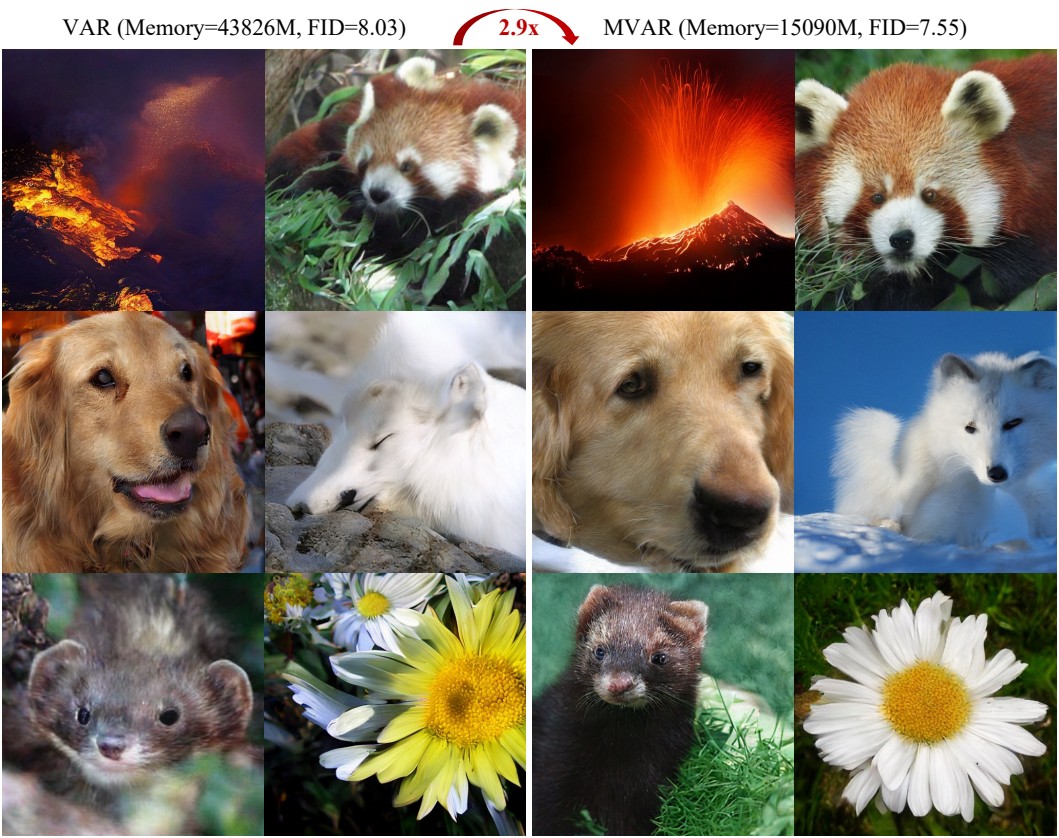

Figure 8: **Qualitative comparison** of VAR-$d16$ and MVAR-$d16$ on the ImageNet 512×512.

## D    MORE EXPERIMENTAL RESULTS

### D.1    EXPERIMENTS ON IMAGENET-512×512

To further evaluate the effectiveness of MVAR, we conducted additional experiments on ImageNet-512×512. Due to limited computational resources and time constraints (VAR was originally trained using a 2.3B model on 256 A100 GPUs with a batch size of 768), we were unable to exactly replicate the experimental settings in the original VAR. Nevertheless, to ensure a fair comparison, we adopted the same training configurations for both VAR and MVAR, training each for 100 epochs with a batch size of 24. For the predefined scale lengths inherited from VAR (1, 2, 3, 4, 6, 9, 13, 18, 24, 32), we parallelized training across scales $r_1$−$r_6$ to maximize GPU utilization, employing the same diagonal-pattern causal masks as described above. Other settings followed the configuration of MVAR-$d20$, as listed in Tab. 5. As shown in Tab. 6 and Fig. 8, MVAR consistently outperforms VAR in generation quality. Moreover, MVAR achieves substantial reductions in memory consumption and training/inference time, demonstrating both its effectiveness and efficiency at higher resolutions.

### D.2    EXPERIMENTS WITH THE 2B MODEL ON CELEBA

In Section 4.2, we reported the performance of MVAR on ImageNet-256×256 using models ranging from 310M to 1B parameters. To further evaluate the scalability of MVAR, we conducted experiments

with a 2B-parameter model. Given limited computational resources (the original VAR was trained on 256 A100 GPUs with a batch size of 1024 for approximately four days), we trained both MVAR and VAR from scratch for 50 epochs on CelebA-256×256 (Liu et al., 2015), using a batch size of 24 distributed across eight RTX 4090 GPUs. The CelebA dataset contains 30,000 images, from which we randomly selected 28,000 for training. As shown in Tab. 7, we report training and inference results on CelebA. MVAR consistently outperforms VAR throughout training. Notably, MVAR achieves a **3.1×** reduction in memory consumption while also enabling faster training and inference.

Table 7: **Quantitative results of MVAR and VAR models (2B) on CelebA-256×256**. Experiments were performed on an RTX 4090 GPU, using a batch size of 64 for inference and 6 or 12 for training.

| Method | Memory↓ | KV Cache↓ | Time↓ | GFLOPs↓ | Train Speed↓ | Train Memory↓ | FID↓ |
|---|---|---|---|---|---|---|---|
| VAR-$d30$ | 46592M | 40108M | 2.94s | 258.60 | 0.56s | 48478M / OOM | 2.65 |
| MVAR-$d30$ | 14804M (**3.1x**) | 0 | 2.35s | 229.89 | 0.36s | 47860M / 48134M | 1.33 |

### D.3 EXPERIMENTS ON CELEBA AND FFHQ DATASETS

To further evaluate the generalization ability of MVAR on additional datasets, we conducted experiments on CelebA (Liu et al., 2015) and FFHQ (Karras et al., 2019). Specifically, MVAR was trained on CelebA at resolutions of 256×256 and 512×512, using the same dataset split strategy as in Section D.2. For FFHQ-256×256, we randomly selected 66,000 images for training and used the remaining images for validation. To ensure fairness and comparability, we adopted identical training settings for VAR and MVAR. The batch sizes were 24 for CelebA-256×256, 6 for CelebA-512×512, and 48 for FFHQ-256×256, with all models trained for 50 epochs. As shown in Tab. 8, MVAR consistently outperforms VAR across both datasets, indicating its strong generalization ability across different data distributions.

Table 8: **Quantitative results of MVAR and VAR on CelebA and FFHQ**. The notation "(k)" denotes the results obtained at the $k$-th epoch.

| Dataset | Method | FID (10)↓ | FID (20)↓ | FID (30)↓ | FID (40)↓ | FID (50)↓ |
|---|---|---|---|---|---|---|
| CelebA | VAR-$d16$ | 4.71 | 4.67 | 3.79 | 3.31 | 3.17 |
|  | MVAR-$d16$ | 4.32 | 4.15 | 3.39 | 3.05 | 2.94 |
| FFHQ | VAR-$d16$ | 3.93 | 3.44 | 2.92 | 2.75 | 2.71 |
|  | MVAR-$d16$ | 3.33 | 2.92 | 2.70 | 2.52 | 2.42 |
| CelebA-512 | VAR-$d16$ | 5.40 | 4.88 | 4.67 | 4.53 | 4.45 |
|  | MVAR-$d16$ | 5.10 | 4.53 | 3.99 | 3.95 | 3.83 |

### D.4 EXPERIMENTS ON NEIGHBORHOOD SIZE $k$ ON CELEBA AND FFHQ

In Section 4.3, we examined the effect of the neighborhood size $k$ in Spatial-Markov attention on ImageNet-256×256, observing diminishing returns but without conducting a comprehensive robustness analysis (e.g., across datasets or image resolutions). Here, we extend this study by evaluating the impact of neighborhood size $k$ on CelebA and FFHQ at different resolutions. Specifically, experiments were performed on CelebA at 256×256 and 512×512, and on FFHQ at 256×256, using the same settings as in Tab. 8. As shown in Tab. 9, larger neighborhoods (e.g., $9 \times 9$) increase computational cost but provide diminishing returns and may even degrade performance. These trends are consistent across datasets and resolutions, aligning with our observations on ImageNet. Overall, the findings suggest that model performance is not highly sensitive to the choice of $k$, thereby mitigating concerns regarding hyperparameter sensitivity.

### D.5 EXPERIMENTS ON TRAINING RATIO IN THE MIXED TRAINING STRATEGY

In Section 3, we introduced the mixed training strategy designed to reduce memory consumption. Due to the strict cross-scale dependency in VAR, where predicting the current scale requires all preceding scales, it is necessary to feed $r_1$ through $r_{10}$ (a total token length of 680) simultaneously with a causal attention mask (see Fig. 5(a)) to model $p(r_l \mid r_{<l})$. In contrast, our approach conditions

Table 9: **Ablation study on neighborhood size** $k$ on CelebA at 256×256 and 512×512 resolutions, and on FFHQ at 256×256 resolution. The notation "(k)" denotes the results obtained at the $k$-th epoch.

| $k$ | Dataset | FID (10) ↓ | FID (20) ↓ | FID (30) ↓ | FID (40) ↓ | FID (50) ↓ | GFLOPs↓ |
|---|---|---|---|---|---|---|---|
| | CelebA | 4.71 | 4.67 | 3.79 | 3.31 | 3.17 | 43.61 |
| – | FFHQ | 3.93 | 3.44 | 2.92 | 2.75 | 2.71 | 43.61 |
| | CelebA-512 | 5.40 | 4.88 | 4.67 | 4.53 | 4.45 | 219.88 |
| | CelebA | 4.47 | 4.26 | 3.42 | 3.23 | 2.99 | 35.11 |
| 5×5 | FFHQ | 3.53 | 3.18 | 2.86 | 2.68 | 2.46 | 35.11 |
| | CelebA-512 | 5.15 | 4.76 | 4.12 | 4.06 | 3.94 | 114.72 |
| | CelebA | 4.32 | 4.15 | 3.39 | 3.05 | 2.94 | 35.44 |
| 7×7 | FFHQ | 3.33 | 2.92 | 2.70 | 2.52 | 2.42 | 35.44 |
| | CelebA-512 | 5.10 | 4.53 | 3.99 | 3.95 | 3.83 | 116.37 |
| | CelebA | 4.41 | 4.24 | 3.52 | 3.21 | 3.04 | 35.89 |
| 9×9 | FFHQ | 4.99 | 2.99 | 2.91 | 2.54 | 2.44 | 35.89 |
| | CelebA-512 | 5.13 | 4.89 | 4.72 | 4.35 | 3.94 | 118.56 |

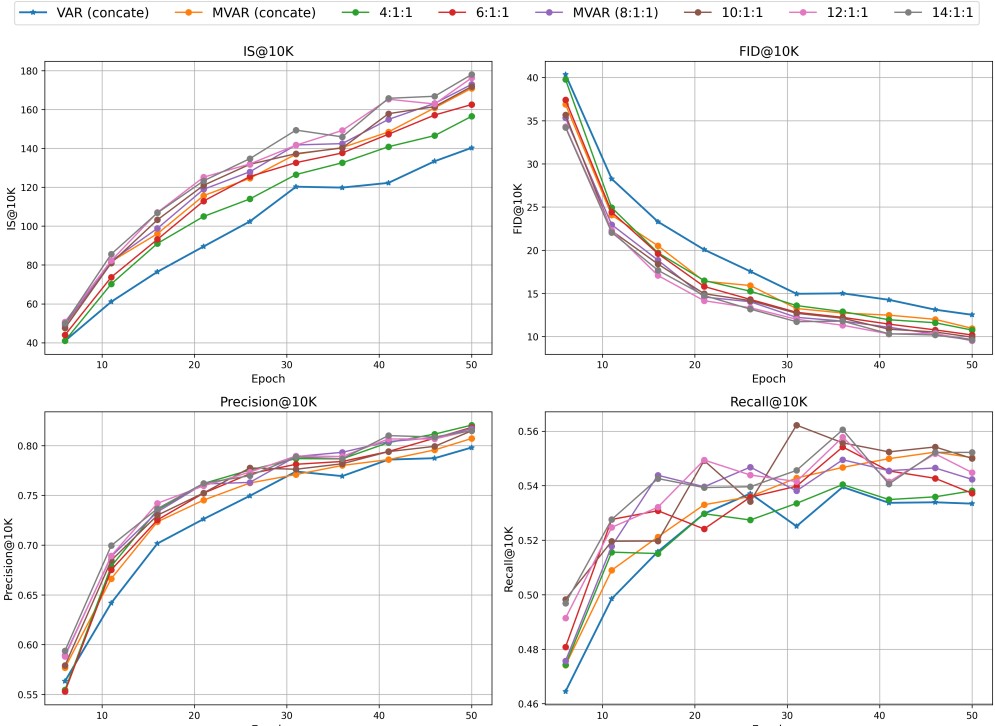

Figure 9: **Ablation study on the training ratio for mixed training at 256×256 generation resolution.** We compare different ratios for training $r_1 - r_8$, $r_9$, and $r_{10}$. Results show that large training ratios (e.g., $8:1:1$) maintain generation quality while substantially reducing GPU memory requirements, demonstrating the robustness and efficiency of the proposed mixed training strategy. The notation "concate" denotes the concatenated training scheme in VAR, "k:1:1" denotes that, within every $k+2$ training iterations, the first $k$ iterations train $r_1 - r_8$, followed by one iteration training $r_9$ and one iteration training $r_{10}$.

only on the immediately preceding scale, which enables a parallel training scheme. Specifically, for 256×256 image generation, we train scales $r_1$ through $r_8$ in parallel using *diagonal-pattern* causal masks (see Fig. 5(c)) that ensure each $r_l$ attends solely to its prefix $r_{l-1}$. For scales $r_9$ and $r_{10}$, which together account for 60% of all tokens, we train them separately using custom CUDA kernels to model $p(r_l \mid \eta_k(r_{l-1}))$. In fact, MVAR can also be trained using VAR's concatenated training scheme by replacing the causal mask in Fig. 5(a) with that in Fig. 5(c). Here, we investigate how the training ratio among $r_1 - r_8$, $r_9$, and $r_{10}$ influences the mixed training strategy. We evaluate the MVAR-$d12$ model on ImageNet at a resolution of 256×256. For fairness, we adopt the same training settings as VAR-$d12$: all models are trained for 50 epochs with a batch size of 64. As shown in Fig. 9,

MVAR consistently outperforms VAR under both the mixed and concatenated training strategies. Moreover, mixed training with larger ratios (e.g., $8\!:\!1\!:\!1$) achieves generation quality comparable to the concatenated training scheme while requiring significantly less GPU memory than VAR. Overall, these results indicate that model performance is not highly sensitive to the choice of larger training ratios, alleviating concerns regarding performance instability under the mixed training strategy.

# E  THE USE OF LARGE LANGUAGE MODELS (LLMS)

We used large language models (LLMs) to aid in polishing the writing. Specifically, LLMs were employed to improve grammar, clarity, and readability of the manuscript. No part of the research ideation, methodological design, or experimental analysis relied on LLMs.

# F  MORE QUALITATIVE RESULTS

- In Fig. 10, we present generation results of our MVAR model on ImageNet at $256{\times}256$ resolution. MVAR produces high-quality images with strong diversity and fidelity.

- In Fig. 11, we compare the performance of VAR with scale and spatial Markovian constants. Our MVAR demonstrates that, even when strong scale and spatial dependencies are removed, it achieves superior results while requiring less GPU memory.

- In Fig. 12, Fig. 13, and Fig. 14, we compare the performance of VAR and MVAR across models of 310M, 600M, and 1B parameters. Our MVAR not only achieves comparable or superior results but also reduces GPU memory consumption by approximately threefold.

- In Fig. 15, we show the attention distributions of VAR together with intermediate outputs at different scales. As the scale increases, the model progressively filters out irrelevant information and concentrates on critical regions in adjacent scales.

- In Fig. 16, Fig. 17, and Fig. 18, we evaluate VAR and MVAR on image in-painting, out-painting, and class-conditional editing tasks. MVAR attains zero-shot performance on par with VAR, while simultaneously reducing GPU memory consumption.

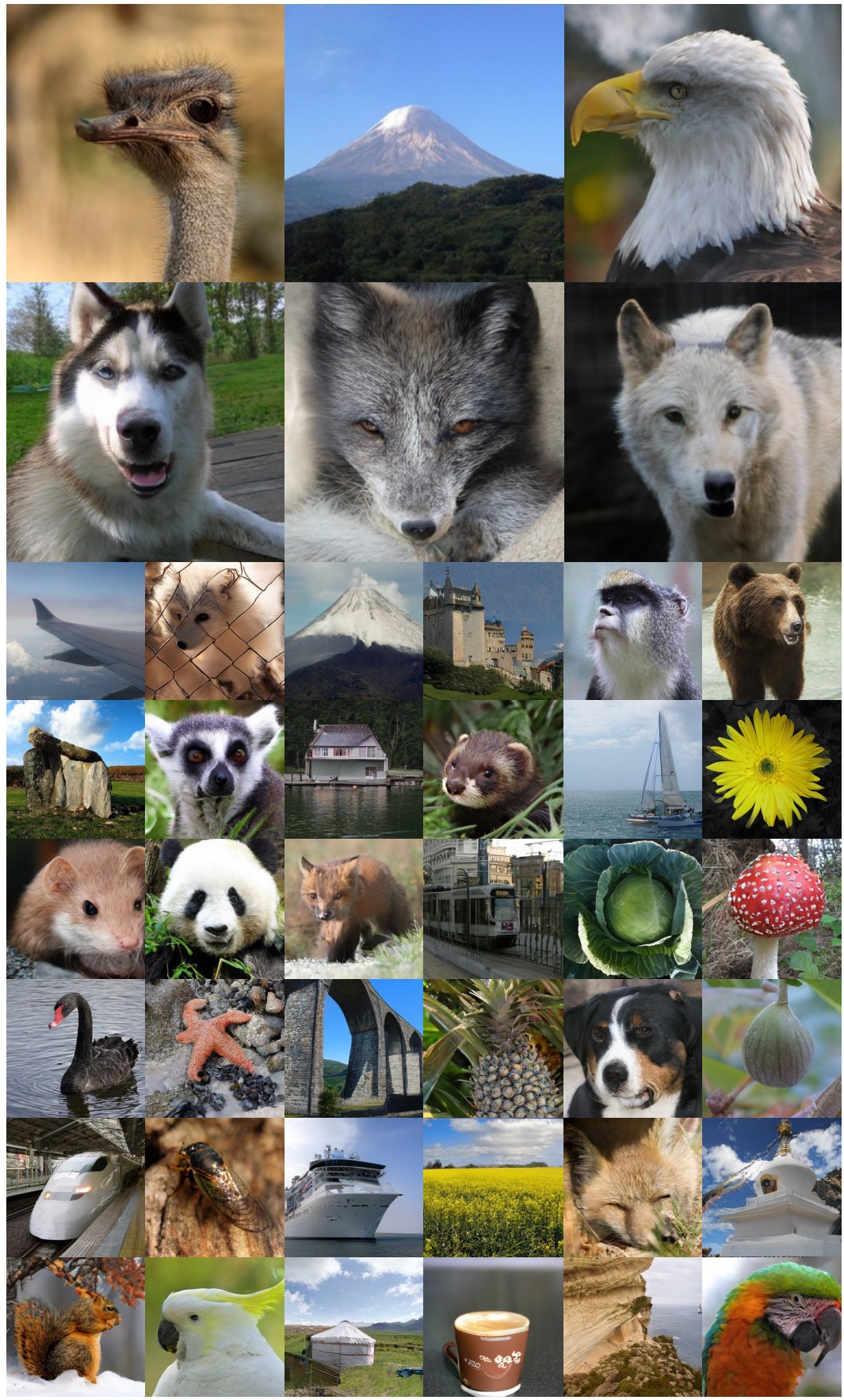

Figure 10: **Qualitative results.** Examples of class-conditional generation on ImageNet 256×256.

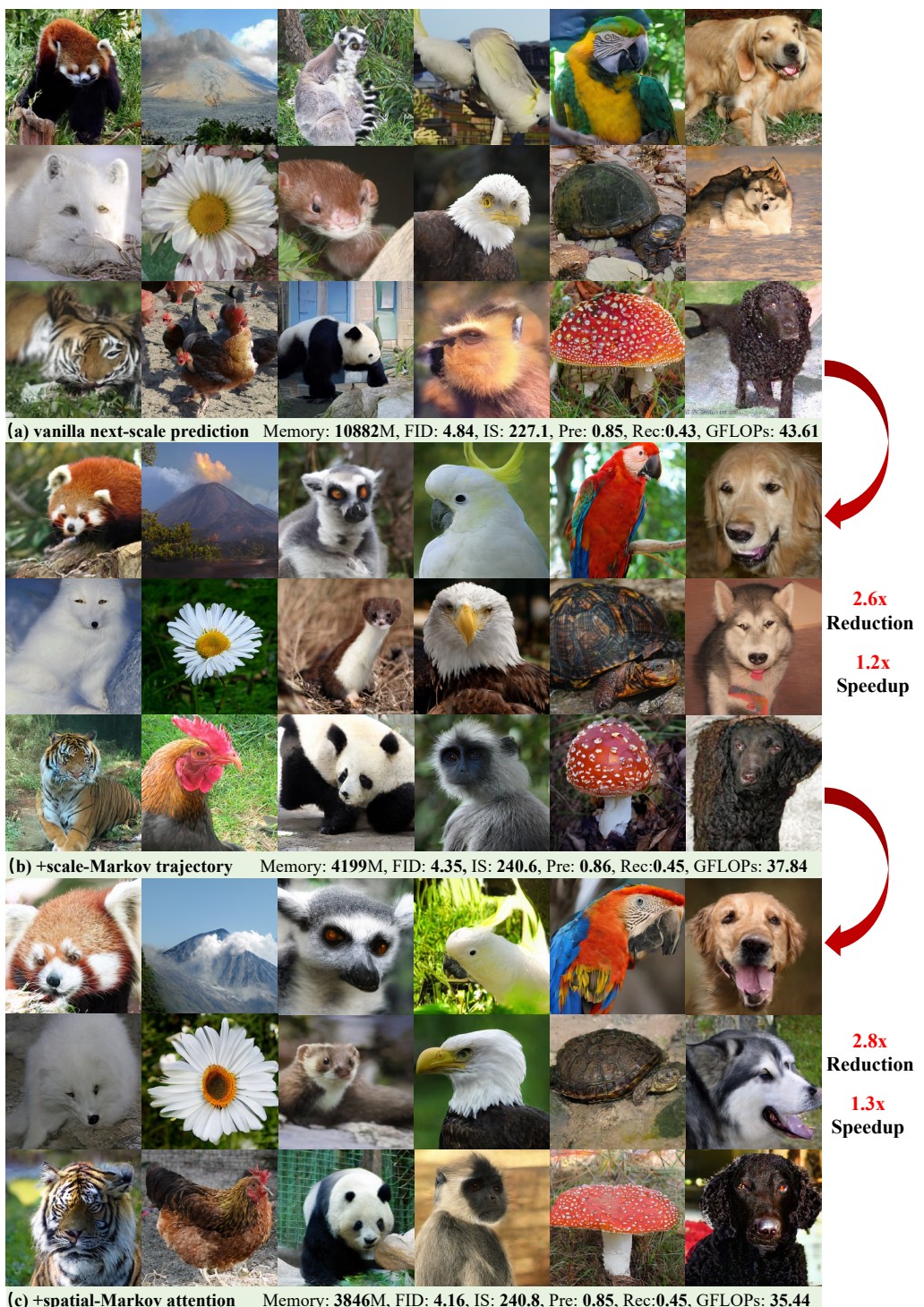

Figure 11: **More qualitative results on scale and spatial Markovian conditioning.** MVAR achieves superior visual quality and requires less GPU memory, even when strong dependencies are removed.

VAR-d16 (Memory=**10882M**, FID=3.55)   **2.8x**   MVAR-d16 (Memory=**3846M**, FID=3.09)

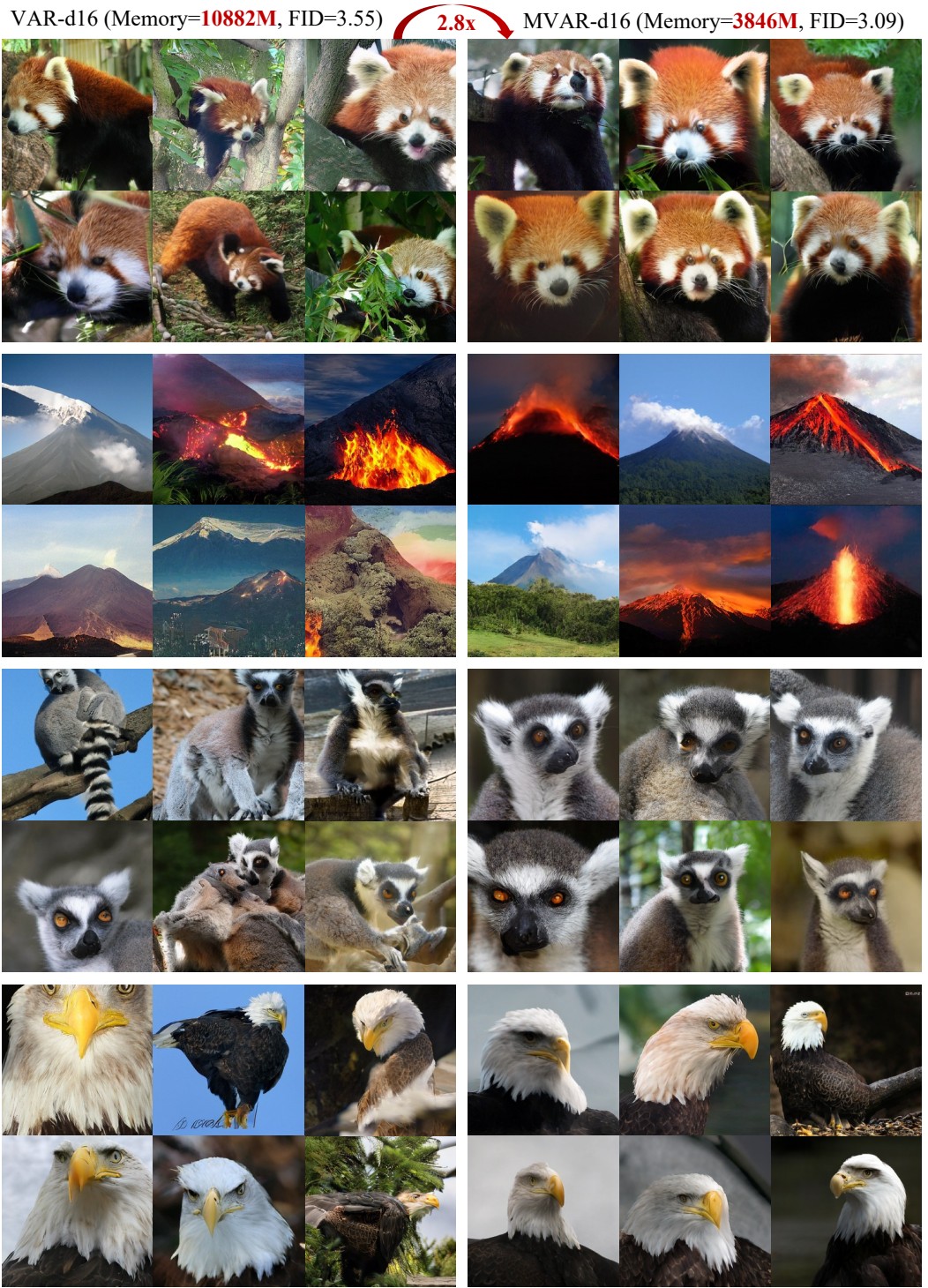

Figure 12: **Qualitative comparison** of VAR-$d$16 and MVAR-$d$16 on the ImageNet 256×256.

VAR-d20 (Memory=**16244M**, FID=2.95)   **3.0x**   MVAR-d20 (Memory=**5432M**, FID=2.87)

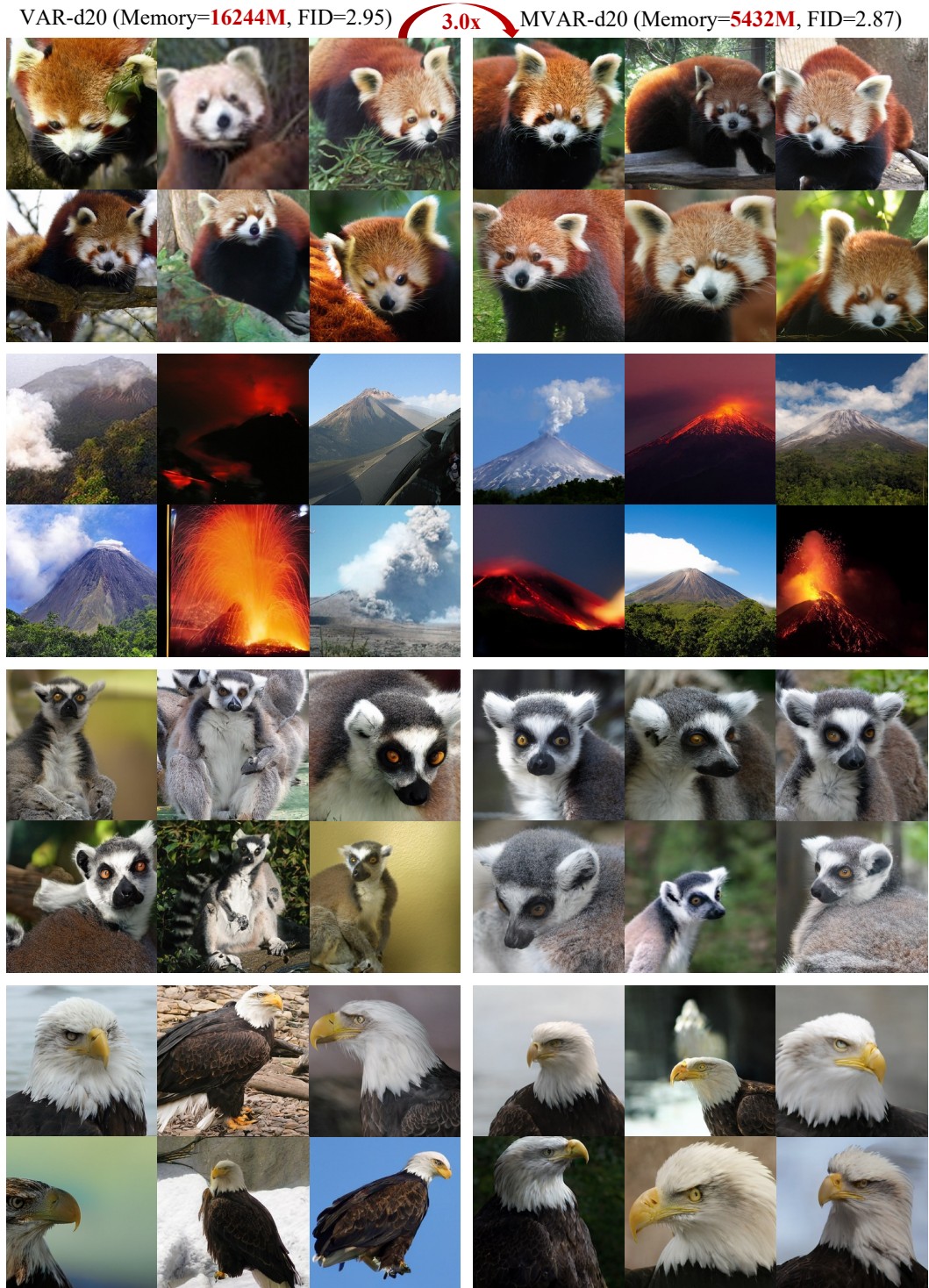

Figure 13: **Qualitative comparison** of VAR-$d$20 and MVAR-$d$20 on the ImageNet 256×256.

VAR-d24 (Memory=**23056M**, FID=2.33)    **3.2x**    MVAR-d24 (Memory=**7216M**, FID=2.23)

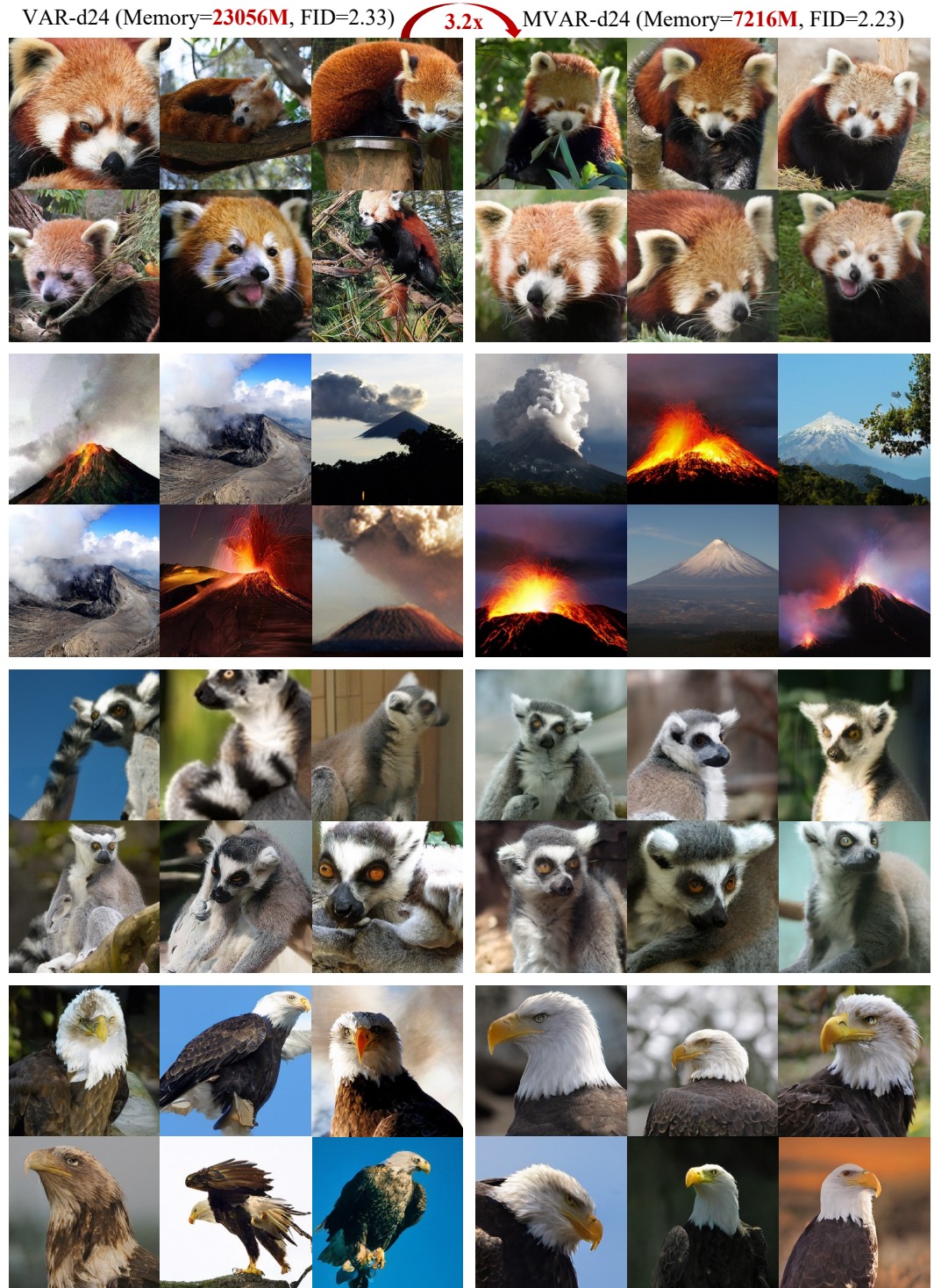

Figure 14: **Qualitative comparison** of VAR-$d24$ and MVAR-$d24$ on the ImageNet 256×256.

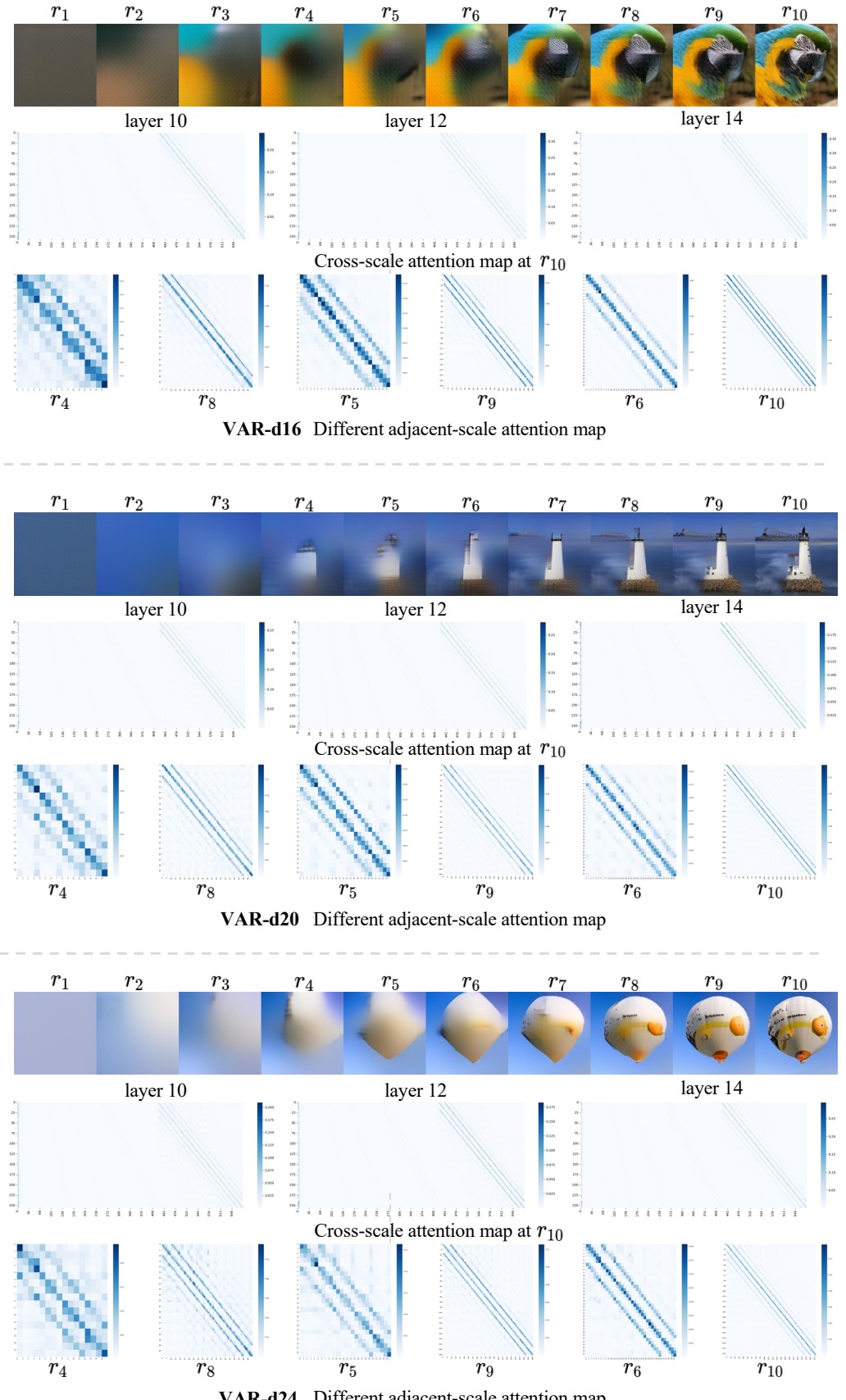

Figure 15: **Attention patterns in vanilla next-scale prediction** (*e.g.*, VAR). Vanilla next-scale prediction exhibits redundant inter-scale dependencies and spatial redundancy.

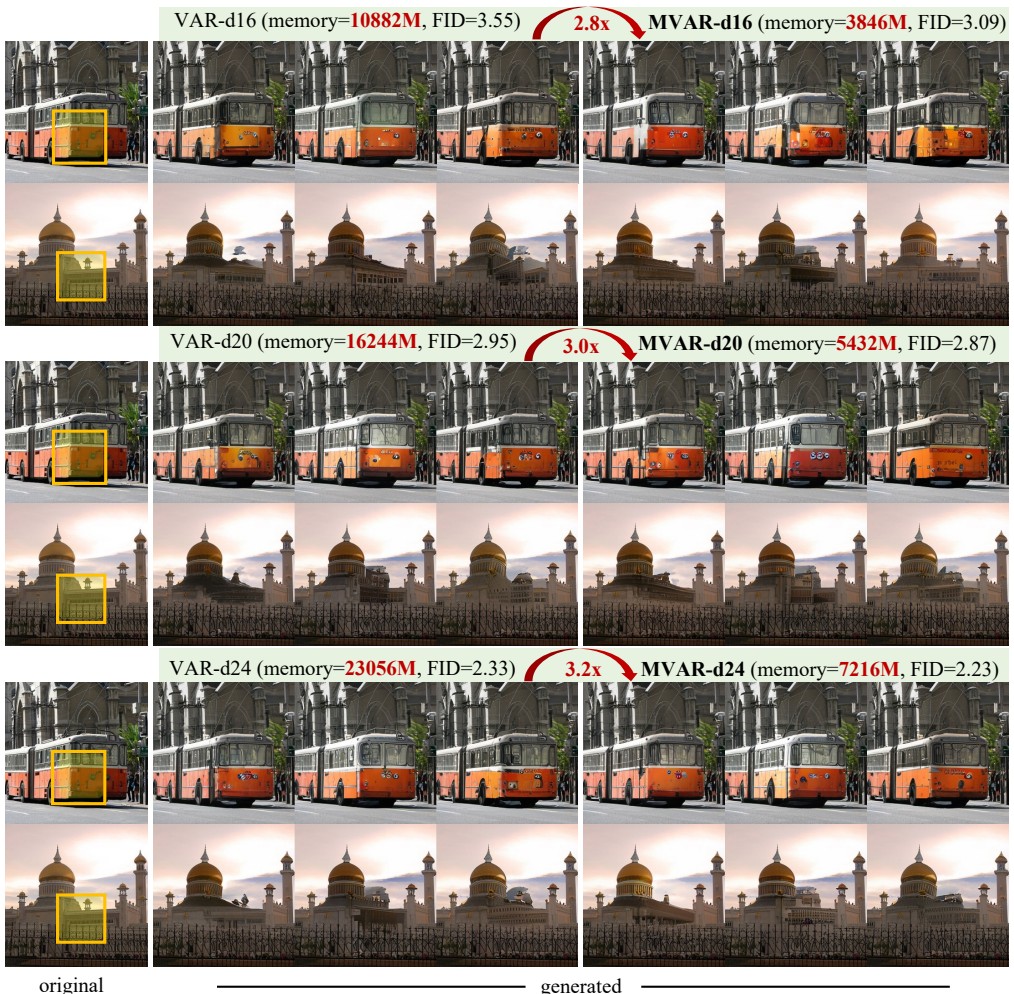

Figure 16: **Zero-shot evaluation in in-painting**. The results show that MVAR can generalize to novel downstream tasks without special design and finetuning. Zoom in for a better view.

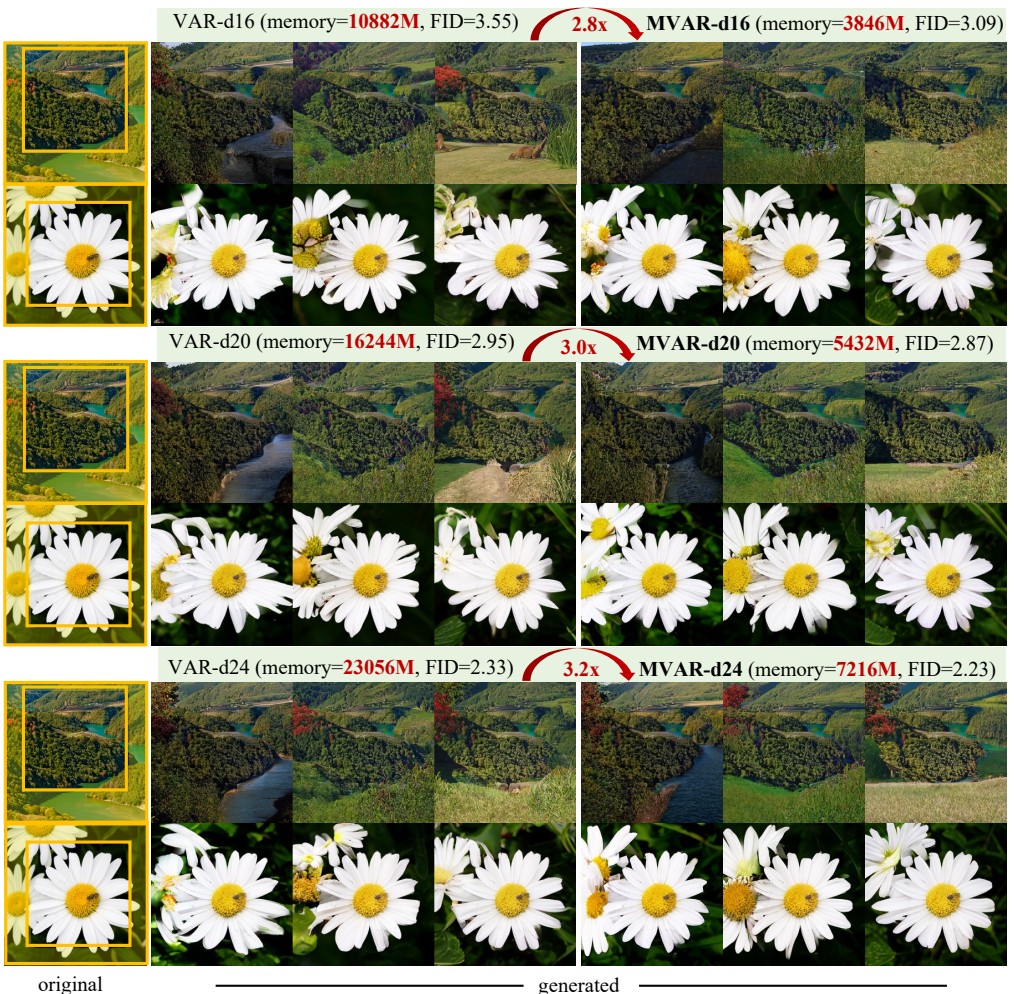

Figure 17: **Zero-shot evaluation in out-painting**. The results show that MVAR can generalize to novel downstream tasks without special design and finetuning. Zoom in for a better view.

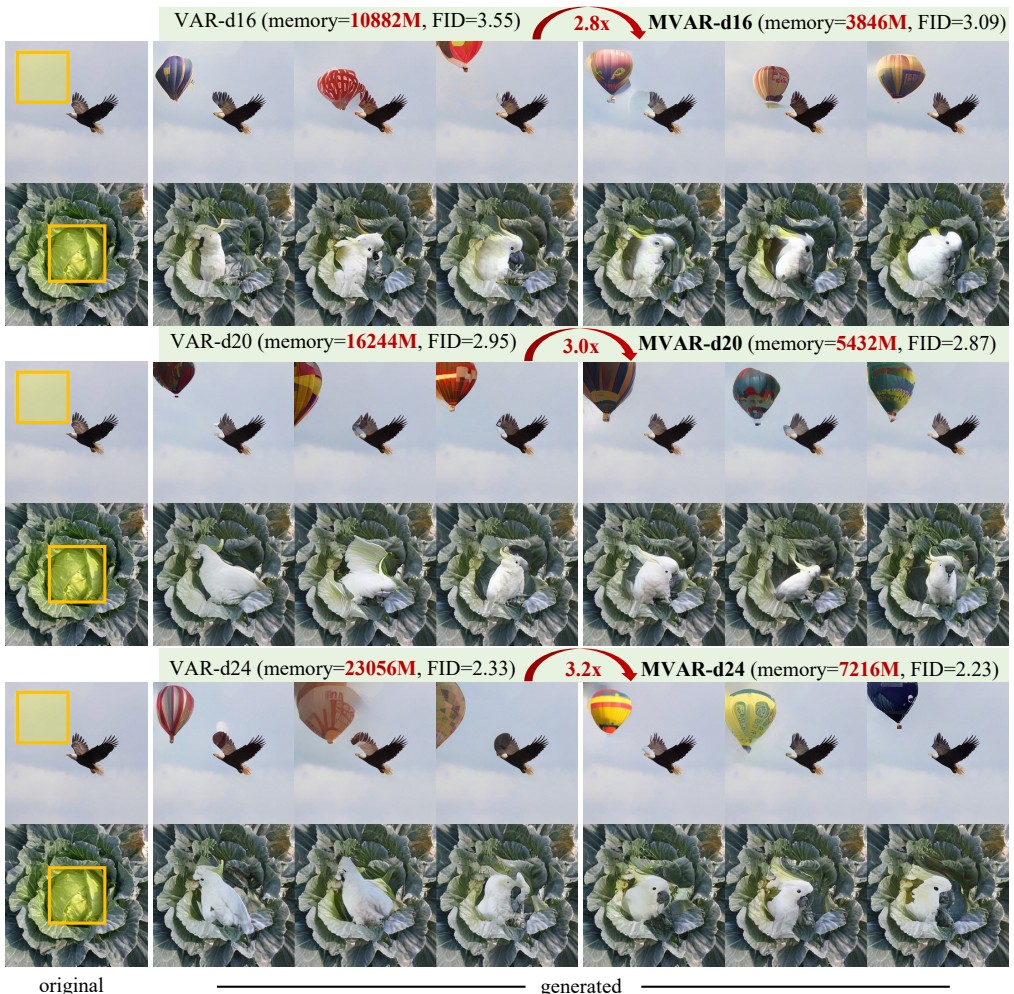

Figure 18: **Zero-shot evaluation in class-conditional editing**. The results show that VAR can generalize to novel downstream tasks without special design and finetuning. Zoom in for a better view.

