# OpenReview forum: "MVAR: Visual Autoregressive Modeling with Scale and Spatial Markovian Conditioning"
_ICLR.cc/2026/Conference — ICLR 2026 Poster_

### Official Review · Reviewer_TLfe · 2025-10-28

**Soundness:** 2
**Presentation:** 3
**Contribution:** 2
**Rating:** 4
**Confidence:** 5

**Summary:**

This paper proposes Markovian Visual AutoRegressive modeling (MVAR). Unlike conventional next-scale prediction methods that condition each scale on all previous scales and require each token to attend to all preceding tokens, MVAR introduces scale and spatial Markovian assumptions to reduce redundancy and computational complexity. Specifically, the model only conditions on adjacent scales (scale-Markov) and restricts attention to local neighborhoods (spatial-Markov), significantly lowering GPU memory usage. Experiments on ImageNet demonstrate that MVAR achieves comparable or superior performance to existing methods while reducing memory consumption by up to 3-4$\times$.

**Strengths:**

- This work introduces Markovian properties into VAR generation process, making the proposed MVAR achieves more efficient training and eliminates the need for a KV cache.

- The paper is well-written and easy to follow.

**Weaknesses:**

- In the proposed spatial Markov property, it seems that the authors restrict each pixel $i$ to interact only with its neighboring pixels within the same scale. Although the term “Markov” is used, performing attention within local neighborhoods has already been extensively explored in previous works.

- In the visualization of attention maps in Figure 3, the behavior of the attention maps produced by this paper appears quite different from those of prior works on VAR like FastVAR[1]. In previous work, the visualized attention maps exhibit strong cross-scale interactions in their attention visualizations. I would like to know how the authors generated Figure 3.

- In Figure 7(b), the attention mask may be incorrect, the current figure seems to consider only intra-scale token interactions while ignoring tokens from the previous scale.

- The paper lacks comparisons with other recent AR-based approaches, such as MAR[2]. Including these comparisons would better situate the proposed MVAR within the broader context of image generation methods.

> [1]FastVAR: Linear Visual Autoregressive Modeling via Cached Token Pruning. ICCV25

> [2]Autoregressive Image Generation without Vector Quantization. NIPS24

**Questions:**

- How was the visualization in Figure 2 obtained? Which dataset and how many samples were used?

- Have the authors designed CUDA kernels for the spatial Markov attention? Does the current implementation affect inference efficiency?

---

> ### Author Response · Authors · 2025-11-19
> **Response to Reviewer TLfe by the Authors of MVAR (Part 1)**
>
> Dear Reviewer TLfe,
>
> Thank you for your positive evaluation and insightful questions. We are pleased that you recognize the value of incorporating Markovian properties into the VAR generation process in our work. We have carefully considered your feedback and revised the manuscript accordingly. Below, we provide detailed responses to your questions.
>
> ------
>
> > **W1: Is the proposed spatial Markov property equivalent to standard local attention ?**
>
> Thank you for pointing this out. While localized attention has been explored in prior work (e.g., the Swin Transformer), these methods are largely driven by architectural considerations and do not explicitly formulate locality from a probabilistic modeling perspective. In contrast, we are the first to study a next-scale autoregressive generative model that uses attention within local neighborhoods from the viewpoint of decomposing the difficulty of modeling the autoregressive likelihood.
>
> Based on the empirical attention patterns of VAR (Figure. 2–3), we observe that both cross-scale and spatial dependencies are already highly localized: tokens at scale $r_l$ attend almost exclusively to scale $r_{l-1}$, and tokens at position $i$ attend primarily to a small $k\times k$ neighborhood. Motivated by these observations, we reformulate the AR likelihood itself:
>
> $p(r_l \mid r_{1:l-1}) \approx p(r_l \mid r_{l-1}), \qquad$
>
> $p(r_l(i)\mid r_{l-1}(\text{all})) \approx p(\eta_k(r_{l-1}(i)).$
>
> This constitutes **scale-Markov and spatial-Markov conditional independence**, analogous to Markov random fields on a 2D grid.
>
> In contrast, prior localized-attention architectures impose local windows for efficiency but do *not* alter the underlying generative factorization or claim such conditional independencies. By enforcing these Markovian assumptions, MVAR not only reduces attention complexity from $\mathcal{O}(N²)$ to $\mathcal{O}(Nk)$ and eliminates KV cache, but also enables per-scale parallel training—benefits not achievable with localized attention alone. Thus, the term “spatial Markovian conditioning” is acceptable, as it describes a probabilistic modeling assumption rather than a simple architectural locality constraint.
>
> ------
>
> > **W2: How Figure 3 was generated?**
>
> First, FastVAR visualizes attention maps at the level of **individual heads**, whereas our MVAR visualizes the mean attention weights aggregated **across all layers and all heads**. Specifically, we first use VAR to generate the ImageNet validation set, producing one image for each of the 1,000 classes. We then visualize the mean attention weights aggregated across all layers and heads at the final scale $r_{10}$.
>
> Second, while some heads in FastVAR indeed show cross-scale interactions, examples such as “Scale10_Head0’’ and “Scale16_Head16’’ exhibit attention behaviors largely independent of earlier scales. Beyond these qualitative observations, our **quantitative analyses** in Figures 2(a) further demonstrate that the VAR model exhibits only weak cross-scale dependencies overall.
>
> We have updated the corresponding visualization details in Appendix Section B for completeness.
>
> ------
>
> > **W3: Figure 7(b) may incorrectly depict attention.**
>
> We believe that the attention mask is correct and provide further clarification below. VAR is trained with ***teacher forcing***, meaning that the input at scale $r_l$ is constructed by **upsampling** the ground-truth outputs of all previous scales $r_{<l}$ to the spatial resolution of $r_l$.
>
> As illustrated in Figure 4 of VAR, the model takes $([s],\, r_1,\, r_2,\, \ldots,\, r_{l-1})$  as input and predicts $(\hat r_1,\, \hat r_2,\, \ldots,\, \hat r_l)$, where each $r_i$ has already been upsampled to match the resolution of $\hat r_{i+1}$. For example, VAR  describes the hierarchical conditioning as:
>
> $p(\hat r_1) = p(r_1 \mid [s]), \quad \hat r_1 \in \mathbb{R}^{1 \times 1}, \quad [s] \in \mathbb{R}^{1 \times 1},$
>
> $p(\hat r_2) = p(r_2 \mid [s], r_1), \quad \hat r_2 \in \mathbb{R}^{2 \times 2}, \quad r_1 \in \mathbb{R}^{2 \times 2},$
>
> $p(\hat r_3) = p(r_3 \mid [s], r_1, r_2), \quad \hat r_3 \in \mathbb{R}^{3 \times 3}, \quad r_2 \in \mathbb{R}^{3 \times 3}.$
>
> Because the inputs to scale $r_l$ already include *upsampled* representations from all previous scales, tokens at scale $r_l$ naturally attend to corresponding upsampled tokens from scale $r_{l-1}$. Therefore, the diagonal-pattern attention mask shown in Figure 5(b) correctly reflects this cross-scale interaction.

---

> ### Author Response · Authors · 2025-11-19
> **Response to Reviewer TLfe by the Authors of MVAR (Part 2)**
>
> > **W4: Lacks comparisons with MAR.**
>
>
> Following the reviewer’s suggestion, we have additionally included comparisons with MAR in Table 1. Under a comparable number of parameters, although our MVAR yields slightly lower generation quality than MAR, it is **68.7×** faster. Moreover, MVAR-d24 attains FID scores comparable to MAR (2.31 for MAR vs. 2.23 for MVAR) while achieving a **26.1×** faster generation speed, highlighting the efficiency advantage of MVAR.
>
> Table 1. Performance comparison between MVAR and MAR.
>
> | Methods  |   TimesTime (sec/image)↓   | FID↓ |  IS↑   | Precision↑ | Recall↑ |
> | --------  | :-------: | :--: | :---: | :--: | :--: |
> | MAR-B      | 18.55 | 2.31 | 281.7 | 0.82 | 0.57 |
> | **MVAR-d16**   |  **0.27 (68.7x)** | 3.09 | 285.5 | 0.85 | 0.51 |
> | **MVAR-d24**    |  **0.71 (26.1x)** | **2.23** | **300.1** | **0.85** | 0.56 |
>
> ------
>
> > **Q1: How Figure 2 was visualized?**
>
>
> For Figure 2(a), we first use VAR to generate the **ImageNet** validation set, producing one image for each of the **1,000** classes. We then compute the mean attention weights across all heads and layers at scale $r_l$ with respect to the preceding scales $r_1, r_2, \ldots, r_{l-1}$. Specifically, for a given scale $r_l$, the attention weights are computed as:
>
> $$
> \textit{Attention}(Q^{l}, K^{l}, V^{l}) = \textit{SoftMax}\left( Q^{l}(K^{<l})^{T} / \sqrt{d}\right) V^{<l}.
> $$
>
> We compute the mean values of
>
> $$
> \frac{\textit{SoftMax}\left( Q^{l}(K^{1})^{T} / \sqrt{d}\right) V^{1}}{N_{1}^2}, \quad
> \frac{\textit{SoftMax}\left( Q^{l}(K^{2})^{T} / \sqrt{d}\right) V^{2}}{N_{2}^2}, \quad \ldots, \quad
> \frac{\textit{SoftMax}\left( Q^{l}(K^{l-1})^{T} / \sqrt{d}\right) V^{l-1}}{N_{l-1}^2},
> $$
> for visualization purposes.
>
> For Figure 2(b), we aggregate the attention weights from  $p\bigl(r_{l} \mid r_{l-1}\bigr)$ ($\textit{SoftMax}\left( Q^{l}(K^{l-1})^{T} / \sqrt{d}\right) V^{l-1}$) at each position $(i,j)$ and average them across different neighborhood sizes $k \times k$.
> We have updated the corresponding visualization details in Appendix Section B for completeness.
>
> ------
>
> > **Q2: Question about the design of CUDA kernels.**
>
> We implement spatial-Markov attention using two alternative approaches: 1) Integration with PyTorch FlexAttention. 2) Custom CUDA kernel implementation based on existing libraries.
>
> As shown in Table 2, compared with the vanilla implementation that uses the attention mask in Figure 5(c), the custom CUDA kernels not only accelerate inference and reduce memory consumption but also improve training speed and memory efficiency.
>
> Table 2. Performance comparison of spatial Markov attention between MVAR, implemented with a CUDA kernel, and MVAR, implemented with an attention mask in Figure 5(c).
>
>
> |        Methods        | TImes (sec/image)↓ |   GFLOPs↓   |  Memory↓  | Train Speed (sec/iter)↓ | Train Memory↓ |
> | :-------------------: | :---------------: | :--------: | :------: | :--------------------: | :----------: |
> |        VAR-d16        |       0.34        |   43.61    |  10882M   |          0.99          |    34319M     |
> | MVAR-d16 (w/o kernel) |       0.29        |   37.84    |   3863M   |          0.72          |    21043M     |
> |     **MVAR-d16**      |     **0.27**      | **35.44**  | **3846M** |        **0.61**        |  **20676M**   |
> |        VAR-d20        |       0.52        |   81.52    |  16244M   |          1.35          |     48173M     |
> | MVAR-d20 (w/o kernel) |       0.48        |   72.50    |   5453M   |          1.21          |    29062M    |
> |     **MVAR-d20**      |     **0.45**      | **68.75**  | **5432M** |        **0.79**        |  **27665M**   |
> |        VAR-d24        |       0.81        |   136.63   |  23056M   |           --           |     OOM      |
> | MVAR-d24 (w/o kernel) |       0.75        |   123.66   |   7397M   |          1.36          |    39476M    |
> |     **MVAR-d24**      |     **0.71**      | **118.25** | **7216M** |        **0.91**        |  **38579M**   |
>
>
> ------
>
> We sincerely hope these responses clarify your concerns. If our revisions and explanations address your questions, we kindly request you to consider updating your score. Please feel free to reach out with any further questions. Thank you again for your careful and constructive review.

---

> > ### Comment · Reviewer_TLfe · 2025-11-26
> > **Response to Author Rebuttals**
> >
> > After reading the authors' rebuttal, my concerns have been addressed. So I will upgrade my score.
> > Moreover, could the authors provide more details about the customized cuda kernel implementation?

---

> > > ### Author Response · Authors · 2025-11-26
> > > **Response to Reviewer TLfe**
> > >
> > > Thank you for acknowledging our work. The details of the customized CUDA kernel implementation are as follows. For each scale $r_l$, we reshape the feature map to $(B, H, H_l \times W_l, d)$. For every query token $i$, we only attend to its $k$ spatial neighbors $\{\eta_k^i(j)\}_{j=1}^k$ instead of all $N_l$ tokens. Mathematically, the kernel computes
> > >
> > > $$
> > > S_i^l(j) = \langle Q_i^l, K_{\eta_k^i(j)}^l \rangle,\quad
> > > \alpha_i^l = SoftMax\left(\frac{S_i^l}{\sqrt{d}}\right),\quad
> > > SA_i^l = \sum_{j=1}^k \alpha_i^l(j)\, V_{\eta_k^i(j)}^l,
> > > $$
> > >
> > >
> > > which corresponds exactly to Eqs. (5–7) in the paper. To realize this efficiently, we use a fused CUDA kernel with the following design:
> > >
> > > 1. Each thread block is assigned to a fixed batch–head pair and a contiguous range of query positions.
> > > 2. For each query, the kernel iterates over its $k$ neighbors, whose indices $\eta_k^i(j)$ are computed on the fly from the 2D coordinates and the kernel size.
> > > 3. All $k$ query–key dot products for a query token are computed in registers, and a numerically stable softmax (subtracting the per-token maximum before exponentiation) is applied over these local scores.
> > > 4. The kernel then immediately accumulates the weighted sum of the corresponding values to produce $\mathrm{SA}_i^l$, without materializing the full attention matrix.
> > >
> > > The kernel therefore has $\mathcal{O}(N_l k d)$ time and $\mathcal{O}(k d)$ memory complexity per head, where $k$ denotes the number of spatial neighbors per query, instead of $\mathcal{O}(N_l^2 d)$ for global attention. Queries, keys, and values are stored in a contiguous memory layout $(B, H, H_l, W_l, d)$ so that threads within a warp always read and write consecutive elements, which yields coalesced global memory accesses. Shared memory is used to cache the query vector and loop over the neighbor keys/values, which significantly reduces DRAM traffic.
> > > A very similar implementation can be found in NATTEN, an open-source project that provides infrastructure for multi-dimensional sparse attention methods, built on top of the CUTLASS project and xFormers.

---

### Official Review · Reviewer_x3u9 · 2025-10-29

**Soundness:** 2
**Presentation:** 3
**Contribution:** 4
**Rating:** 6
**Confidence:** 3

**Summary:**

This paper proposes a novel visual autoregressive framework, MVAR, which introduces scale-Markov (depending only on adjacent scales) and spatial-Markov (focusing only on a local neighborhood) assumptions. This approach addresses the significant computational and memory redundancy in existing "next-scale prediction" models, reducing attention complexity from $O(N^2)$ to $O(Nk)$, cutting down GPU memory usage, and enabling an inference process that is entirely free of a KV cache.

**Strengths:**

1. The paper accurately identifies and solves a critical bottleneck in existing VAR models related to memory and computation (especially the KV cache). This is a practical problem that has hindered scaling these models to larger sizes and higher resolutions.
2. MVAR's two core components (scale-Markov trajectory and spatial-Markov attention) are conceptually simple but highly effective. The "KV-cache-free" and "parallel training" properties drastically reduce the training cost.

**Weaknesses:**

1. While the scale-Markov assumption proves effective in the current setup, it might be an approximation. In more complex scenarios requiring strong long-range dependencies (e.g., images with complex global structures or multi-object interactions), discarding all information from $r_1$ to $r_{l-2}$ could become a performance bottleneck.
2. The paper mentions that training for $r_1$ to $r_8$ is parallel, but $r_9$ and $r_{10}$ (which account for 60% of the tokens) seem to be handled separately using single-scale conditional modeling. The complexities of this mixed training strategy and its impact on final performance consistency are not sufficiently discussed in the main text.

**Questions:**

1. The ablation study (Table 3) shows that method (d) achieves a better FID than using all prior scales. Does this imply that the original VAR design (relying on all scales) is not just redundant, but potentially a **harmful inductive bias**? Does it force the model to process irrelevant historical information, thereby interfering with the generation?
2. MVAR uses a fixed $k \times k$ neighborhood. Have you considered using dynamic or deformable neighborhoods? For example, using a larger $k$ in smooth areas of an image and a smaller $k$ in complex, textured regions?
3. Appendix B mentions that the causal mask is disabled for the last two scales. Does this mean that within these scales, token generation is parallel (i.e., non-autoregressive)? If so, does this contradict the model's definition as an "autoregressive" framework? Please clarify how $p(r_l | \eta_k(r_{l-1}))$ is modeled for these final two scales.

---

> ### Author Response · Authors · 2025-11-19
> **Response to Reviewer  x3u9 by the Authors of MVAR (Part 1)**
>
> Dear Reviewer x3u9,
>
> Thank you for your positive evaluation and insightful questions. We are pleased that you recognize the value of incorporating Markovian properties into the VAR generation process in our work. We have carefully considered your feedback and revised the manuscript accordingly. Below, we provide detailed responses to your questions.
>
> ------
>
> > **W1: Will the scale-Markov assumption fail to hold in more complex scenarios?**
>
> Thank you for pointing this out. We believe that this potential issue will not arise in the context of the next-scale generation paradigm.
> **First**, unlike the next-token approach, which generates images in raster-scan order and relies primarily on local semantic information, the next-scale paradigm preserves the complete global structure at each scale. In the next-token approach, if each token depends solely on preceding tokens, global semantic consistency may be compromised, creating a performance bottleneck. By contrast, in the next-scale method, even for images with complex global structures or multi-object interactions, the information from the most recent scale already aggregates all relevant context, making it unnecessary for the current scale to directly depend on earlier ones.
>
> **Second**, both the qualitative analysis in Figure 2(a) and the attention visualization in Figure 3(a) confirm that these cross-scale dependencies are weak in practice. As shown in Figures 7(a) and 7(b), enforcing this dependency constraint during training reduces interference from earlier scales and improves overall generation performance.
>
> ---
>
> > **W2: The complexities of this mixed training strategy and its impact should be discussed.**
>
>
> We clarify that this mixed training strategy is adopted solely to reduce memory consumption.
>
> Due to the strict cross-scale dependency in VAR, predicting the current scale requires all preceding scales. This necessitates feeding $r_1$ through $r_{10}$ (token length 680) simultaneously with a causal attention mask to model $p(r_l \mid r_{<l})$. In contrast, MVAR only conditions on the immediately adjacent preceding scale, enabling a fully parallel training scheme. Following the reviewer’s suggestion, we evaluated a setup consistent with VAR by replacing the causal attention mask in Figure 5(a) with the one in Figure 5(c). As shown in **Table 1**, regardless of whether MVAR adopts the mixed or concatenated training strategy, it consistently outperforms VAR. Moreover, although the concatenated training strategy achieves generation quality comparable to that of the mixed training strategy, it incurs higher memory consumption.
>
> Furthermore, in **Section D.5 of the Appendix in the revised paper**, we investigate the impact of different training ratios between $r_1$–$r_8$, $r_9$, and $r_{10}$ under the mixed training strategy. We implement the MVAR-d12 model at 256×256 resolution on ImageNet, and for fair comparison, follow the same training settings as VAR-d12: all models are trained for 50 epochs with a batch size of 64. As shown in **Figure 9 of the Appendix**, larger training ratios (e.g., 8:1:1) yield generation quality comparable to the concatenated training scheme while consuming significantly less GPU memory than VAR.
>
> Overall, these results indicate that model performance is not highly sensitive to the choice of larger training ratios, alleviating concerns about performance inconsistency under mixed training.
>
> Table 1. Comparison of training strategies, evaluating mixed training against concatenated training.
>
> | Methods                     | Inference Time (sec/image)↓ | GFLOPs↓ | KV Cache↓ | Inference  Memory↓ | Train Speed (sec/iter)↓ | Train Memory↓ | FID↓ | IS↑   | Precision↑ | Recall↑ |
> | :-------------------------: | :-------------------------: | :-----: | :-------: | :----------: | :---------------------: | :-----------: | :--: | :---: | :--------: | :-----: |
> | VAR-d16                     | 0.34                        | 43.61   | 5704M     | 10882M       | 0.99                    | 34319M        | 4.84 | 227.1 | 0.85       | 0.43    |
> | MVAR-d16 (concatenated training) | 0.27                        | 35.44   | 0        | 3846M (2.8x) | 0.99 | 34319M | 4.14 | 241.5 | 0.86 | 0.45 |
> | MVAR-d16 (mixed training)   | 0.27                        | 35.44   | 0         | 3846M (2.8x) | 0.61 (1.6x)             | 20676M        | 4.16 | 240.8 | 0.85       | 0.45    |

---

> ### Author Response · Authors · 2025-11-19
> **Response to Reviewer x3u9 by the Authors of MVAR (Part 2)**
>
> > **Q1: Does the VAR design potentially introduce harmful inductive biases that interfere with the generation process?**
>
> We concur with this observation. **First**, both the qualitative analysis in Figure 2 and the attention visualization in Figure 3 confirm that cross-scale dependencies in VAR and its full-attention mechanism are weak in practice. **Second**, as described in Line 092, introducing this type of Markovian constraint during training encourages the model to focus on the most relevant information, enhancing modeling accuracy and reducing interference from less informative historical scales. As shown in Figure 7, enforcing this dependency constraint reduces interference from earlier scales and irrelevant tokens, thereby improving overall generation performance.
>
> ---
>
> > **Q2: Have author considered using dynamic or deformable neighborhoods?**
>
> Following the reviewer’s suggestion, we conducted experiments using dynamic neighborhoods. As shown in Figure 7, during next-scale generation, the earlier small scales generate smooth structural content, while the later large scales focus on textured regions.
>
> Based on the scale settings (1, 2, 3, 4, 5, 6, 8, 10, 13, 16), we applied neighborhood sizes of 7×7, 5×5, and 3×3 for $r_8$ (10×10), $r_8$ (10×10)  and $r_{10}$ (16×16), respectively. As reported in **Table 2**, using smaller neighborhood sizes at larger scales can also yield better performance than VAR, further validating the presence of spatial redundancy in VAR models.
>
> Table 2. Comparison of neighborhood size choices, evaluating dynamic vs. static settings.
>
>
> |      Methods       | Inference Time (sec/image)↓ | GFLOPs↓ | KV Cache↓ | Inference  Memory↓ | Train Speed (sec/iter)↓ | Train Memory↓ | FID↓ |  IS↑  | Precision↑ | Recall↑ |
> | :----------------: | :-------------------------: | :-----: | :-------: | :----------------: | :---------------------: | :-----------: | :--: | :---: | :--------: | :-----: |
> |      VAR-d16       |            0.34             |  43.61  |   5704M   |       10882M       |          0.99           |    34319M     | 4.84 | 227.1 |    0.85    |  0.43   |
> |      MVAR-d16      |            0.27             |  35.44  |     0     |    3846M (2.8x)    |       0.61 (1.6x)       | 20676M (1.7x) | 4.16 | 240.8 |    0.85    |  0.45   |
> | MVAR-d16 (dynamic) |            0.26             |  34.81  |     0     |    3837M (2.8x)    |       0.53 (1.7x)       | 18253M (1.9x) | 4.32 | 235.6 |    0.84    |  0.44   |
>
> ---
>
> > **Q3:  Please clarify how $p\bigl(r_{l} \mid \eta_{k}(r_{l-1})\bigr)$ is modeled for these final two scales.**
>
> We appreciate the reviewer’s comment and provide further clarification below.
>
> First, we clarify how $p\bigl(r_{l} \mid r_{l-1}\bigr)$ is modeled for the final two scales. VAR is trained with ***teacher forcing***, meaning that the input at scale $r_l$ is constructed by **upsampling** the ground-truth outputs of all previous scales $r_{<l}$ to the spatial resolution of $r_l$. As illustrated in Figure 4 of VAR, the model takes $([s],\, r_1,\, r_2,\, \ldots,\, r_{l-1})$  as input and predicts $(\hat r_1,\, \hat r_2,\, \ldots,\, \hat r_l)$, where each $r_i$ has already been upsampled to match the resolution of $\hat r_{i+1}$.
> For example, VAR  describes the hierarchical conditioning as:
>
> $p(\hat r_1) = p(r_1 \mid [s]), \quad \hat r_1 \in \mathbb{R}^{1 \times 1}, \quad [s] \in \mathbb{R}^{1 \times 1},$
>
> $p(\hat r_2) = p(r_2 \mid [s], r_1), \quad \hat r_2 \in \mathbb{R}^{2 \times 2}, \quad r_1 \in \mathbb{R}^{2 \times 2},$
>
> $p(\hat r_3) = p(r_3 \mid [s], r_1, r_2), \quad \hat r_3 \in \mathbb{R}^{3 \times 3}, \quad r_2 \in \mathbb{R}^{3 \times 3}.$
>
> Because the inputs to scale $r_l$ already include *upsampled* representations from all previous scales, tokens at scale $r_l$ naturally attend to corresponding upsampled tokens from scale $r_{l-1}$. Consequently, $r_9$ and $r_{10}$ can be trained separately to realize $p\bigl(r_{l} \mid r_{l-1}\bigr)$.  Based on this approach, the only remaining component to implement is spatial-Markov attention ($p\bigl(r_{l} \mid \eta_{k}(r_{l-1})\bigr)$ ). We achieve this using PyTorch FlexAttention and a custom CUDA kernel built upon existing libraries.
>
> ---
>
> We sincerely hope these responses clarify your concerns. If our revisions and explanations address your questions, we kindly request you to consider updating your score. Please feel free to reach out with any further questions. Thank you again for your careful and constructive review.

---

> ### Comment · Reviewer_x3u9 · 2025-11-26
>
> The authors have fully addressed my concerns, and I have decided to maintain my rating.

---

> > ### Author Response · Authors · 2025-11-26
> > **Response to Reviewer x3u9**
> >
> > Dear Reviewer x3u9,
> >
> > Thank you for acknowledging our response. We appreciate your valuable feedback and support.
> >
> > Best regards,
> >
> > Paper 3301 Authors

---

### Official Review · Reviewer_3S78 · 2025-11-01

**Soundness:** 3
**Presentation:** 3
**Contribution:** 2
**Rating:** 4
**Confidence:** 4

**Summary:**

This paper proposes Markovian Visual AutoRegressive Modeling (MVAR), an efficient framework for visual autoregressive generation. The method introduces Markovian assumptions in both scale and spatial dimensions: each scale depends only on its immediately preceding scale, and each token attends only to a local neighborhood. This design effectively removes redundant dependencies in next-scale prediction and reduces attention complexity from $O(N^2)$ to $O(Nk)$. Experiments on ImageNet show that MVAR achieves similar or slightly better generation quality compared to vanilla VAR while using significantly less memory and allowing parallel training.

**Strengths:**

- The motivation is clear and well supported by empirical observations of redundancy in next-scale prediction.
- The Markovian formulation is conceptually simple yet brings strong computational benefits.
- The method achieves 3× memory reduction without degrading generation quality.
- Results on ImageNet demonstrate good efficiency–performance trade-offs, making the approach practical for large-scale settings.
- The paper is overall well written and easy to follow.

**Weaknesses:**

- The experimental comparison is limited. The paper does not include results against related methods such as Randomized Autoregressive Visual Generation (RAVG, 2024), which also targets efficiency improvement in visual AR models.
- Table 1 reports results compared with VAR-d16, but it is unclear whether the MVAR model also uses 16 decoder layers. The discrepancy between Table 1 and Table 2 results (different FID/IS scores) suggests inconsistent model settings.
- Several experiments in the appendix only report FID without IS or precision/recall, making the evaluation incomplete.

**Questions:**

See weaknesses.

---

> ### Author Response · Authors · 2025-11-19
> **Response to Reviewer 3S78 by the Authors of MVAR**
>
> Dear Reviewer 3S78,
>
> Thank you for your positive evaluation and insightful questions. We are glad that you recognize the motivation and empirical observations presented in our work. We have carefully considered your feedback and revised the manuscript accordingly. Below, we provide detailed responses to your questions.
>
> ---
>
> > **W1: Lacks comparisons with RAR[1].**
>
> Thank you for pointing this out. RAR [1] introduces a randomized permutation strategy during training, which enables bidirectional context learning while remaining fully compatible with standard language-modeling frameworks, thereby improving generation quality.
>
> Following the reviewer’s suggestion, we have additionally included comparisons with RAR in Table 1. As shown, although RAR improves generation quality through its permutation-based training mechanism, its reliance on next-token prediction leads to substantially slower inference, which limits its practical usability. Moreover, for larger models, RAR incurs considerably higher memory consumption, posing additional disadvantages for real-world deployment.
>
> Table 1. Performance comparison between MVAR and RAR
>
> | Methods  |   Time (sec/image)↓   |  Memory (M)↓  | FID↓ |  IS↑   | Precision↑ | Recall↑ |
> | :------:  | :--------: | :------: | :--: | :---: | :--: | :--: |
> | RAR-B     |    3.35     |   5461    | 1.95 | 290.5 | 0.82 | 0.58 |
> | **MVAR-d16** | **0.27 (12.4x)** | **3846 (1.4x)** | 3.09 | 285.5 | **0.85** | 0.51 |
> | RAR-L     |    3.67     |   7395    | 1.70 | 299.5 | 0.81 | 0.60 |
> | **MVAR-d20** |  **0.45 (8.2x)**  | **5432 (1.4x)** | 2.87 | 295.3 | **0.86** | 0.52 |
> | RAR-XXL    |    8.42     |  17017   | 1.48 | 326.0 | 0.80 | 0.63 |
> | **MVAR-d24** |  **0.71 (11.9x)**  | **7216 (2.4x)** | 2.23 | 300.1 | **0.85** | 0.56 |
>
> ---
>
> > **W2: Discrepancy between Table 1 and Table 2 results.**
>
> We apologize for the confusion. **First**, the letter “d’’ denotes the depth of the Transformer in the autoregressive network, rather than the number of decoder layers. This clarification has been explicitly added to Table 1 in the revised manuscript. **Second**, the results in Table 1 are obtained by **training MVAR-d16 from scratch**, whereas the results in Table 2 are obtained by **fine-tuning** MVAR-d16 from pretrained VAR-16 weights. This difference in training protocol explains the discrepancy in reported FID/IS scores.
>
> The fine-tuning setting serves two purposes: 1) it demonstrates that existing VAR models contain both scale-level and spatial redundancies, which can be effectively reduced with minimal computational cost while maintaining comparable performance and lowering memory usage;  2) it offers a more resource-efficient alternative, as training large VAR models from scratch would otherwise require more than 16 A100 GPUs.
>
> ---
>
> > **W3: Only the FID is reported in the Appendix.**
>
> We followed the common evaluation protocol used in VQ-GAN[2], where FID is reported on CelebA and FFHQ, and FID, IS, Precision, and Recall are reported on ImageNet. Following the reviewer’s suggestion, we have updated **Tables 7, 8, and 9 in the Appendix** in the revised manuscript to include the full set of metrics. As shown, MVAR consistently achieves superior performance over VAR across all reported measures.
>
> ------
>
> We sincerely hope these responses clarify your concerns. If our revisions and explanations address your questions, we kindly request you to consider updating your score. Please feel free to reach out with any further questions. Thank you again for your careful and constructive review.
>
> ---
>
> > **References**
>
> [1] Yu, Qihang, et al. "Randomized autoregressive visual generation." *Proceedings of the IEEE/CVF International Conference on Computer Vision*. 2025.
>
> [2] Esser, Patrick, Robin Rombach, and Bjorn Ommer. "Taming transformers for high-resolution image synthesis." *Proceedings of the IEEE/CVF conference on computer vision and pattern recognition*. 2021.

---

### Official Review · Reviewer_u1aY · 2025-11-07

**Soundness:** 4
**Presentation:** 3
**Contribution:** 3
**Rating:** 6
**Confidence:** 4

**Summary:**

This paper introduces MVAR, which exploits scale and spatial redundancy based upon Visual Autoregressive Modeling (VAR). Specifically, MVAR only uses the preceding scale for next-scale prediction, and restricts the attention map of each token to a localized neighborhood. These two changes significantly reduces the computational complexity from O(N^2) to O(NK), and eliminate the need for KV cache during inference.

**Strengths:**

1. This paper is overall well written and rather easy to follow. The figures look nice and captures the core idea in a glimpse.

2. The scale and spatial redundancy in original VAR work makes intuitive sense to me, and the proposed solution is both straightforward and effective as shown in experiments.

3. MVAR shows both performance advantages over vanilla VAR, as well as reduced memory footprint and accelerated inference speed.

**Weaknesses:**

1. Only model size of 300M is studied in this paper. It is unclear if MVAR shows good scalability or not. The authors are encouraged to show the scaling trend following VAR (up to 2B model).

2. Beyonds ImageNet unconditional generation, The authors are encouraged to try the image in-painting and out-painting task and class-conditional image editing task as in the zero-shot setup in VAR paper.

**Questions:**

1. Based on Table 1, the introduction of markovian scale prediction and localized attention leads to even better performance (FID 3.09 vs 3.55, IS 285.5 vs 280.4)? Would the authors care to explain thsi phenomenon?

2. And how much performance gain does scale markovian conditioning along brings?

3. Why is it called "spatial Markovian conditioning"? Localized attention is a common operation, dating back to the days of Swin-Transformer. I do not see any markovian part in localized attention.

---

> ### Author Response · Authors · 2025-11-19
> **Response to Reviewer u1aY by the Authors of MVAR**
>
> Dear Reviewer u1aY,
>
> Thank you for your positive evaluation and insightful questions. We are pleased that you recognize the value of incorporating Markovian properties into the VAR generation process in our work. We have carefully considered your feedback and revised the manuscript accordingly. Below, we provide detailed responses to your questions.
>
> ------
>
> > **W1:  The authors are encouraged to show the scaling trend following VAR.**
>
> Thank you for pointing this out. We have implemented a 2B-parameter model (MVAR-d30) to assess scalability in **Table 7 of Appendix**. Due to limited computational resources and time constraints (the original VAR-d30 was trained on **256 A100 GPUs with a batch size of 1024** for approximately four days), we trained both MVAR-d30 and VAR-d30 from scratch for 50 epochs on CelebA, using the same settings. The results in Table 7 show that MVAR-d30 consistently outperforms VAR-d30 throughout training. Notably, MVAR-d30 achieves a 3× reduction in memory consumption while also offering faster training and inference speeds.
>
> ---
>
> > **W2: The authors are encouraged to present the zero-shot setup in VAR.**
>
> Following the reviewer’s suggestion, we have added results for the image in-painting, out-painting, and class-conditional image editing tasks in the revised paper. As shown in **Figures 16, 17, and 18 of the Appendix E**, our MVAR attains zero-shot performance on par with VAR, while simultaneously reducing GPU memory consumption.
>
> ---
>
> > **Q1: Could the authors explain why introducing the Markovian property can improve performance?**
>
> **First**, both the qualitative analysis in Figure 2 and the attention visualization in Figure 3 confirm that cross-scale dependencies in VAR and its full-attention mechanism are weak in practice. **Second,** as described in Line 092, introducing this type of Markovian constraint during training encourages the model to focus on the most relevant information, enhancing modeling accuracy and reducing interference from less informative historical scales.
>
> As shown in Figure 7, enforcing this dependency constraint reduces interference from earlier scales and irrelevant tokens, thereby improving overall generation performance.
>
> ---
>
> > **Q2: How much performance gain does scale markovian conditioning along brings?**
>
> As shown in Table 3 (a) and (d), as well as Figures 7 and 11, incorporating the scale-Markov trajectory not only accelerates inference (from 0.34 sec/image to 0.29 sec/image, with a 2.6× reduction in memory) and training (from 0.99 sec/iteration to 0.58 sec/iteration, with a 1.7× reduction in memory) but also improves generation quality (FID from 4.84 to 4.35).
>
> ---
>
> > **Q3: Why is it called "spatial Markovian conditioning"?**
>
> Thank you for pointing this out. While localized attention has been explored in prior work (e.g., the Swin Transformer), these methods are largely driven by architectural considerations and do not explicitly formulate locality from a probabilistic modeling perspective. In contrast, we are the first to study a next-scale autoregressive generative model that uses attention within local neighborhoods from the viewpoint of decomposing the difficulty of modeling the autoregressive likelihood.
>
> Based on the empirical attention patterns of VAR (Figure. 2–3), we observe that both cross-scale and spatial dependencies are already highly localized: tokens at scale $r_l$ attend almost exclusively to scale $r_{l-1}$, and tokens at position $i$ attend primarily to a small $k\times k$ neighborhood. Motivated by these observations, we reformulate the AR likelihood itself:
>
> $p(r_l \mid r_{1:l-1}) \approx p(r_l \mid r_{l-1}), \qquad$
>
> $p(r_l(i)\mid r_{l-1}(\text{all})) \approx p(\eta_k(r_{l-1}(i)).$
>
> This constitutes **scale-Markov and spatial-Markov conditional independence**, analogous to Markov random fields on a 2D grid.
>
> In contrast, prior localized-attention architectures impose local windows for efficiency but do *not* alter the underlying generative factorization or claim such conditional independencies. By enforcing these Markovian assumptions, MVAR not only reduces attention complexity from $\mathcal{O}(N²)$ to $\mathcal{O}(Nk)$ and eliminates KV cache, but also enables per-scale parallel training—benefits not achievable with localized attention alone. Thus, the term “spatial Markovian conditioning” is acceptable, as it describes a probabilistic modeling assumption rather than a simple architectural locality constraint.
>
> ---
>
> We sincerely hope these responses clarify your concerns. If our revisions and explanations address your questions, we kindly request you to consider updating your score. Please feel free to reach out with any further questions. Thank you again for your careful and constructive review.

---

> ### Comment · Reviewer_u1aY · 2025-11-25
>
> The authors have adequately addressed my concerns. I tend to keep my rating as marginally above the acceptance threshold.

---

> > ### Author Response · Authors · 2025-11-25
> > **Response to Reviewer u1aY**
> >
> > Dear Reviewer u1aY,
> >
> > Thank you for acknowledging our response. We appreciate your valuable feedback and support.
> >
> > Best regards,
> >
> > Paper 3301 Authors

---

### Author Response · Authors · 2025-11-19
**Response to All Reviewers by the Authors of MVAR**

We thank all reviewers for their time, insightful suggestions, and valuable comments. Below, we provide detailed responses to each reviewer’s feedback and have revised the main paper and appendix accordingly. The main changes are summarized as follows:

- We provide a detailed explanation of why the diagonal-pattern mask in Figure 5(b) corresponds to modeling the conditional distribution $p(r_l \mid r_{l-1})$,  addressing Reviewer TLfe’s comment in W3 and x3u9’s comment in Q3.

- We added comparison results with MAR and RAR, as requested by Reviewer TLfe in W4 and Reviewer 3S78 in W1.

- We provide a more detailed description of how Figures 2 and 3 are visualized in Section B of the Appendix, in response to Reviewer TLfe's comments in W2 and Q1.

- We added Appendix D.5, titled "Experiments on Training Ratio in the Mixed Training Strategy," to illustrate the influence of the mixed training strategy, as requested by Reviewer x3u9 in W2.

- We added Figures 16, 17, and 18 in Appendix E to demonstrate that our method achieves zero-shot performance comparable to VAR, as requested by Reviewer u1aY in W2.


We hope that these revisions adequately address the reviewers’ concerns. We sincerely thank the reviewers again for their valuable feedback.

Best regards,
Paper 3301 Authors

---

### Meta-Review · Area_Chair_dF4P · 2026-01-06

**Summary:**

AC thinks that most of the concerns have been addressed, and sees some novelty in the approach. MVAR should be an useful technique to report to the community and the AC recommends accept.

In particular, the reviewers all agreed that the method to assume markovian to predict the next scale has novelty and the results are promising. In addition, the reviewers are all aligned on the efficiency boost provided by the paper's method.

**Reviewer Concerns:**

Most have been addressed.

**Reviewer Scores:**

AC thinks that most if not all the reviewers would have stayed or increased to 6.

---

### Decision · Program_Chairs · 2026-01-26

Accept (Poster)